

# Riemannian optimization of photonic quantum circuits in phase and Fock space

**Yuan Yao[1,2,3]⋆, Filippo Miatto[1,2,3]† and Nicolás Quesada[4]‡**

**1** Télécom Paris, LTCI, 20 Place Marguerite Perey, 91120 Palaiseau, France
**2** Institut Polytechnique de Paris, 5 Av. Le Chatelier, 91764 Palaiseau, France
**3** Xanadu, 777 Bay St. M5G 2C8 Toronto, ON, Canada
**4** Department of Engineering Physics, École Polytechnique de Montréal,
Montréal, QC, H3T 1J4, Canada

⋆ yuan.yao@telecom-paris.fr , † filippo@xanadu.ai , ‡ nicolas.quesada@polymtl.ca

## Abstract

We propose a framework to design and optimize generic photonic quantum circuits composed of Gaussian objects (pure and mixed Gaussian states, Gaussian unitaries, Gaussian channels, Gaussian measurements) as well as non-Gaussian effects such as photon-number-resolving measurements. In this framework, we parametrize a phase space representation of Gaussian objects using elements of the symplectic group (or the unitary or orthogonal group in special cases), and then we transform it into the Fock representation using a single linear recurrence relation that computes the Fock amplitudes of any Gaussian object recursively. We also compute the gradient of the Fock amplitudes with respect to phase space parameters by differentiating through the recurrence relation. We can then use Riemannian optimization on the symplectic group to optimize $M$-mode Gaussian objects, avoiding the need to commit to particular realizations in terms of fundamental gates. This allows us to "mod out" all the different gate-level implementations of the same circuit, which now can be chosen after the optimization has completed. This can be especially useful when looking to answer general questions, such as bounding the value of a property over a class of states or transformations, or when one would like to worry about hardware constraints separately from the circuit optimization step. Finally, we make our framework extendable to non-Gaussian objects that can be written as linear combinations of Gaussian ones, by explicitly computing the change in global phase when states undergo Gaussian transformations. We implemented all of these methods in the freely available open-source library MrMustard [1], which we use in three examples to optimize the 216-mode interferometer in Borealis, and 2- and 3-modes circuits (with Fock measurements) to produce cat states and cubic phase states.

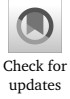

# 1 Introduction

Gaussian quantum mechanics [2–4] is a subset of quantum mechanics that finds applications in several fields of quantum physics, such as quantum optics [5], quantum key distribution [6], optomechanical systems [7], quantum chemistry [8], condensed matter systems [9]. In the context of quantum optics, many of the available states (e.g. coherent, squeezed, thermal), transformations (e.g. beam splitter, squeezer, displacement, attenuator, amplifier), and measurements (e.g. homodyne, heterodyne) are Gaussian, i.e. characterized by a Gaussian Wigner function. Gaussian objects are easy to simulate, but in order to access a broader (in fact, universal) set of states and transformations, one needs to include non-Gaussian effects. One way to take into account non-Gaussian effects (e.g. photon-number-resolving detection), is to transform from the Gaussian phase space representation to the Fock space representation. Hence, studying the Fock space representation of Gaussian objects plays an important role in optical quantum simulation and optical quantum information processing [10–15].

The Fock space representation of Gaussian objects has been studied in different communities: in chemical physics, one studies vibronic transitions using the Hermite polynomials as a computational tool [16–19], and the matrix elements of unitary Gaussian and non-Gaussian transformations have been evaluated in [20] by using the multimode Bogoliubov transformation. In the mathematical physics context, these transformations correspond to the Bargmann-Fock representation of the symplectic group (also known as a metaplectic representation or oscillator representation), which we can understand as the Fock space representation of the group of Gaussian transformations [21].

In [22], we introduced a method to compute the Fock space amplitudes of Gaussian unitary transformations using a generating function. Part of the present work presents a unified picture of all the Gaussian objects covering pure states, mixed states, and Gaussian channels as well. While many libraries exist to simulate quantum optical circuits [23–30], and some of them have automatic differentiation capabilities, ours is the first one to fully exploit the properties of Gaussian quantum mechanics in Fock- and Phase-space while being differentiable. Thus, we implemented all of the methods and algorithms derived here in the open-source library MrMustard [1]. We showcase out methods by performing optimization on photonic circuits directly. This allows us to generate interesting non-Gaussian states [31].

Besides an open-source library, we present three results, significantly extending what we did in [22]: (i) In Sec. 3 we introduce a unified, differentiable recursive representation of pure [11,32] and mixed Gaussian states [10,12,33], Gaussian unitaries [34,35] and Gaussian channels in Fock space. We emphasize that while the results for states and unitary channels were already known in the literature, the results for channels, are to the best of our knowledge, presented here for the first time. (ii) In Sec. 4 we compute the global phase of the composition of Gaussian operations, which allows our method to be extended to states and transformations beyond the Gaussian ones as proposed in [36]. (iii) In Sec. 5 we show how to perform a Riemannian optimization of $M$-modes Gaussian objects directly on the underlying symmetry group, bypassing the need to decompose them into some arrangement of fundamental elements and therefore allowing us to optimize a Gaussian quantum circuit as an entire block. Our method first focuses on the optimization of the Gaussian objects and their associated symplectic matrices. We illustrate the utility of our methods and library by finding simple Gaussian circuits for the heralded preparation of cat states with mean photon number 4, fidelity 99.38%, and success probability 7.39%. Similarly, we obtain cubic-phase states with cubic-phase gate parameter $\gamma = 0.3\hbar$ and squeezing $r = -1$, fidelity 99.00%, and success probability 0.06%. Finally, we also benchmark our Riemannian optimization method showing that in non-Gaussian state preparation tasks it converges faster and with smaller variance than Euclidean optimization over parameters of gate-decomposed circuits.

We will adopt the following notation conventions. The transposition and Hermitian conjugation operations are denoted as $\cdot^T$ and $\cdot^\dagger$. We use boldface for vectors $\boldsymbol{r}$ and matrices $\boldsymbol{S}$ but denote their components as $r_i$ and $S_{ij}$ respectively. We use $0_M$ for the $M \times M$ null matrix, $\boldsymbol{0}$ for a zero vector, and $0$ for a scalar zero. The single-mode vacuum state denotes as $|0\rangle$ and the multimode vacuum state is $|\boldsymbol{0}\rangle$. $\mathbb{1}_M$ denotes for the $M \times M$ identity matrix. Given a vector of integers $\boldsymbol{n} = (n_1, \ldots, n_M)$ we write $\boldsymbol{n}! = \prod_{i=1}^M n_i!$, $|\boldsymbol{n}\rangle = |n_1\rangle \otimes |n_2\rangle \otimes \ldots \otimes |n_M\rangle$ and given a complex or real vector $\boldsymbol{\alpha} = (\alpha_1, \ldots, \alpha_M)^T$ we write $\boldsymbol{\alpha}^{\boldsymbol{n}} = \prod_{i=1}^M \alpha_i^{n_i}$ and $\partial_{\boldsymbol{\alpha}}^{\boldsymbol{n}} = \prod_{i=1}^M \frac{\partial^{n_i}}{\partial \alpha_i^{n_i}}$. We write H.c. for Hermitian conjugate term.

## 2 Gaussian formalism

### 2.1 Commutation relations

Given an $M$-mode quantum continuous variable system, the field operators (i.e. annihilation and creation operators) $a_j$, $a_j^\dagger$; $j \in \{1, 2, \ldots, M\}$ satisfy the canonical commutation relation [37]:

$$[a_i, a_j^\dagger] = \delta_{ij}, \quad [a_i, a_j] = [a_i^\dagger, a_j^\dagger] = 0. \tag{1}$$

We can express these relations in a compact way by defining a vector of annihilation and creation operators $\boldsymbol{z} = (a_1, \ldots, a_M, a_1^\dagger, \ldots, a_M^\dagger)$, so that we can write

$$[z_i, z_j^\dagger] = Z_{ij}, \tag{2}$$

with

$$\boldsymbol{Z} = \begin{pmatrix} \mathbb{1}_M & 0_M \\ 0_M & -\mathbb{1}_M \end{pmatrix}. \tag{3}$$

An alternative way to describe continuous-variable systems is obtained by defining the hermitian position $q$ and momentum $p$ operators:

$$q_j = \sqrt{\tfrac{\hbar}{2}}(a_j^\dagger + a_j), \quad p_j = i\sqrt{\tfrac{\hbar}{2}}(a_j^\dagger - a_j). \tag{4}$$

We can group these operators into a quadrature vector $\boldsymbol{r} = (q_1, \ldots, q_M, p_1, \ldots, p_M)$ so that $\boldsymbol{r}$ is related to $\boldsymbol{z}$ by the unitary matrix $\boldsymbol{W}$:

$$\boldsymbol{r} = \sqrt{\hbar}\boldsymbol{W}\boldsymbol{z}, \tag{5}$$

where

$$\boldsymbol{W} = \frac{1}{\sqrt{2}} \begin{pmatrix} \mathbb{1}_M & \mathbb{1}_M \\ -i\mathbb{1}_M & i\mathbb{1}_M \end{pmatrix}, \tag{6}$$

and $i = \sqrt{-1}$ is the imaginary unit.

Combining Eq. (2) and Eq. (5), we have:

$$[r_j, r_k] = \hbar(\boldsymbol{W}^\dagger \boldsymbol{Z} \boldsymbol{W})_{jk} = i\hbar\Omega_{jk}, \tag{7}$$

where $\boldsymbol{\Omega}$ is the skew-symmetric matrix:

$$\boldsymbol{\Omega} = \begin{pmatrix} 0_M & \mathbb{1}_M \\ -\mathbb{1}_M & 0_M \end{pmatrix} = \begin{pmatrix} 0 & 1 \\ -1 & 0 \end{pmatrix} \otimes \mathbb{1}_M, \tag{8}$$

which is central to the description of the symplectic group (see section 5). A brief summary of properties of the symplectic group can be found in Appendix A.

## 2.2 Gaussian states

A Gaussian state is any state whose characteristic functions and quasi-probability distributions are Gaussian functions in phase space [37]. Some well-known examples are coherent states, squeezed states, thermal states, and the vacuum state (which is the only state which is at the same time Gaussian and a number eigenstate).

The characteristic function of a state with density matrix $\rho$ is defined as:

$$\chi(\boldsymbol{s};\rho) = \text{Tr}(\mathcal{D}_{\boldsymbol{s}}\rho), \tag{9}$$

where $\mathcal{D}_{\boldsymbol{s}} = \exp(-i\boldsymbol{s}^T\boldsymbol{\Omega}\boldsymbol{r}/\hbar)$ is the Weyl, or displacement, operator and $\boldsymbol{s} \in \mathbb{R}^{2M}$ is a real vector in phase space.

For a Gaussian state we write the characteristic function in terms of its mean vector $\bar{\boldsymbol{r}}$ and covariance matrix $\boldsymbol{V}$ as [38]

$$\chi(\boldsymbol{s};\rho) = \exp\left[-\tfrac{1}{2}\boldsymbol{s}^T\boldsymbol{\Omega}^T\boldsymbol{V}\boldsymbol{\Omega}\boldsymbol{s} - i\bar{\boldsymbol{r}}^T\boldsymbol{\Omega}\boldsymbol{s}\right], \tag{10}$$

where

$$\bar{r}_i = \langle r_i \rangle, \tag{11}$$

$$V_{ij} = \frac{1}{2}\langle r_i r_j + r_j r_i \rangle - \bar{r}_i\bar{r}_j. \tag{12}$$

Note that the covariance matrix $\boldsymbol{V}$ is a real, symmetric, positive definite matrix.

If we use the amplitude basis $\boldsymbol{z}$, we find the mean vector $\bar{\boldsymbol{\mu}}$ and the covariance matrix $\boldsymbol{\sigma}$:

$$\bar{\mu}_i = \langle z_i \rangle = \frac{1}{\sqrt{\hbar}}\left(\boldsymbol{W}^\dagger\bar{\boldsymbol{r}}\right)_i, \tag{13}$$

$$\sigma_{ij} = \frac{1}{2}\langle z_i z_j^\dagger + z_j z_i^\dagger \rangle - \bar{\mu}_i\bar{\mu}_j^\dagger = \frac{1}{\hbar}(\boldsymbol{W}^\dagger\boldsymbol{V}\boldsymbol{W})_{ij}. \tag{14}$$

Compared with the real covariance matrix $\boldsymbol{V}$, we denote the $\boldsymbol{\sigma}$ as the complex covariance matrix.

In the remainder of this paper we will write the phase space description of a Gaussian state as the pair $(\boldsymbol{V}, \bar{\boldsymbol{r}})$ or $(\boldsymbol{\sigma}, \bar{\boldsymbol{\mu}})$ depending on which basis we use. For example, the vacuum state $|0\rangle$, which satisfies $a_j|0\rangle = 0$, has a zero mean vector and covariance matrix $\boldsymbol{V} = \frac{\hbar}{2}\mathbb{1}_2$ or $\boldsymbol{\sigma} = \frac{1}{2}\mathbb{1}_2$.

## 2.3 Gaussian transformations

Gaussian unitary transformations are those that map Gaussian states to Gaussian states [38], thus in the Schrödinger picture, an input Gaussian state $\rho$ is mapped to an output Gaussian state

$$\rho \mapsto \rho' = U_G\rho U_G^\dagger. \tag{15}$$

Gaussian unitaries have as generators polynomials of at most degree 2 in the quadratures (or equivalently in the creation and annihilation operators).

In the Heisenberg picture a Gaussian unitary (parameterized by a $2M \times 2M$ matrix $\boldsymbol{S}$ and a real vector $\boldsymbol{d}$ with size $2M$) transforms the quadrature operators as follows

$$\boldsymbol{r} \mapsto \boldsymbol{r}' = U_G^\dagger\boldsymbol{r}U_G = \boldsymbol{S}\boldsymbol{r} + \boldsymbol{d}. \tag{16}$$

Since $\boldsymbol{r}'$ is obtained from $\boldsymbol{r}$ by unitary conjugation, it must satisfy the canonical commutation relations in Eq. (7). This implies that the matrix $\boldsymbol{S}$ satisfies

$$\boldsymbol{S}\boldsymbol{\Omega}\boldsymbol{S}^T = \boldsymbol{\Omega}, \tag{17}$$

that is, $S$ must be an element of the (real) symplectic group, $S \in \mathrm{Sp}(2M, \mathbb{R})$.

An $M$-mode Gaussian unitary generated by a second-degree polynomial in the quadratures can be decomposed into an $M$-mode displacement $\mathcal{D}_d$ and an $M$-mode unitary generated by a strictly quadratic unitary that is responsible for the symplectic matrix appearing in Eq. (16) and thus we can write [39]

$$U_G = \mathcal{D}_d U(S), \tag{18}$$

where $\mathcal{D}_d$ is the displacement operator, parametrized by a real vector $d$ of size $2M$. We can also express the $M$-mode displacement operator as the tensor product of the single-mode displacement operator, with a complex vector $\gamma$ of size $M$. The relation between the vector $d$ and $\gamma$ can be derived from Eq. (4):

$$d = \sqrt{2\hbar}[\Re(\gamma), \Im(\gamma)]. \tag{19}$$

The single-mode displacement operator is defined as

$$\mathcal{D}(\gamma) = \exp\left[\gamma a^\dagger - \gamma^* a\right]. \tag{20}$$

We will also give the definitions of other single-mode Gaussian unitaries, noting that the multimode version is just the tensor product extension of their single-mode version.

The single-mode rotation operator

$$\mathcal{R}(\phi) = \exp\left[i\phi a^\dagger a\right], \tag{21}$$

which has $d_{\mathrm{rot}} = 0$ and

$$S_{\mathrm{rot}} = \begin{bmatrix} \cos\phi & -\sin\phi \\ \sin\phi & \cos\phi \end{bmatrix}. \tag{22}$$

The single-mode squeezing operator is defined as

$$\mathcal{S}(\zeta) = \exp\left[\frac{1}{2}\zeta^* a^2 - \mathrm{H.c.}\right], \tag{23}$$

where $\zeta = r e^{i\delta}$, and it has $d_{\mathrm{sq}} = 0$ and

$$S_{\mathrm{sq}} = S_{\mathrm{rot}}(\delta/2) \begin{bmatrix} e^{-r} & 0 \\ 0 & e^r \end{bmatrix} S_{\mathrm{rot}}(\delta/2)^T. \tag{24}$$

An $M$-mode interferometer with Hilbert space operator [34]

$$\mathcal{W}(J) = \exp\left[i \sum_{k,l=1}^{M} J_{k,l} a_k^\dagger a_l\right], \tag{25}$$

which has $d_{\mathrm{intf}} = 0$, and

$$S_{\mathrm{intf}} = \begin{bmatrix} \Re(U) & -\Im(U) \\ \Im(U) & \Re(U) \end{bmatrix}, \tag{26}$$

where $U = \exp[iJ]$ is a unitary matrix (since $J = J^\dagger$).

Note that $S_{\mathrm{intf}} \in \mathrm{Sp}(2n, \mathbb{R}) \cap \mathrm{O}(2n) \cong U(n)$ where $\mathrm{Sp}(2n, \mathbb{R})$ is the symplectic group, $\mathrm{O}(2n)$ is the orthogonal group and $U(n)$ is the unitary group (cf. Appendix B of Serafini [38]).

A particular instance of an interferometer is the one beamsplitter, parametrized in terms of transmission angle $\theta$ and a phase $\phi$ (the energy transmission is given by $\cos^2\theta$). In this case we have

$$J = i \begin{pmatrix} 0 & \theta e^{-i\phi} \\ -\theta e^{i\phi} & 0 \end{pmatrix}, \qquad U = \begin{pmatrix} \cos\theta & -e^{-i\phi}\sin\theta \\ e^{i\phi}\sin\theta & \cos\theta \end{pmatrix}. \tag{27}$$

Note that our definition of interferometer immediately implies that $\mathcal{W}(J)|0\rangle = |0\rangle$ without any ambiguity in the global phase of the state on the right hand side.

Gaussian unitaries transform the mean vector $\bar{r}$ and the covariance matrix $V$ of a Gaussian state as [2]:

$$(V, \bar{r}) \mapsto (V', \bar{r}') = (SVS^T, S\bar{r} + d). \tag{28}$$

A deterministic Gaussian channel is the most general trace-preserving map between Gaussian states. It is characterized by two matrices $X, Y$ and a vector $d$ [38]. The action of the channel on a Gaussian state $(V, \bar{r})$ is

$$(V, \bar{r}) \mapsto (XVX^T + Y, X\bar{r} + d), \tag{29}$$

where the matrices $X$ and $Y$ need to satisfy

$$Y + i\frac{\hbar}{2}\Omega \geq i\frac{\hbar}{2}X\Omega X^T. \tag{30}$$

More generally, the action of a Gaussian channel on the characteristic function of an arbitrary state amounts to

$$\chi(s) \mapsto \chi'(s) = \chi(\Omega^T X^T \Omega s) \exp\left(-\tfrac{1}{2}s^T \Omega^T Y \Omega s - i d^T \Omega s\right). \tag{31}$$

Note that unitary channels such as Eq. (28) are special cases of a Gaussian channels where $Y = 0_{2M}$ and $X$ is symplectic. More generally, when $X$ is not symplectic and thus the channel is not unitary, the matrix $Y$ represents added noise in the state.

Examples of single-mode Gaussian channels are the pure loss channel (defined in Eq.(5.77) in the book [38]) by energy transmission $0 \leq \eta \leq 1$, which has

$$X = \sqrt{\eta}\mathbb{1}_2, \quad Y = \tfrac{\hbar}{2}(1-\eta)\mathbb{1}_2, \quad d = 0, \tag{32}$$

and the amplification channel (defined in Eq.(5.87) in the book [38]) with energy gain $g \geq 1$, which has

$$X = \sqrt{g}\mathbb{1}_2, \quad Y = \tfrac{\hbar}{2}(g-1)\mathbb{1}_2, \quad d = 0. \tag{33}$$

An example of a multi-mode Gaussian channel is the lossy interferometer parametrized in terms of a transmission matrix $T$ with singular values bounded from above by 1. For this channel, we find

$$X = \begin{bmatrix} \Re(T) & -\Im(T) \\ \Im(T) & \Re(T) \end{bmatrix}, \tag{34}$$

$$Y = \tfrac{\hbar}{2}\left(\mathbb{1}_{2M} - XX^T\right), \tag{35}$$

$$d = 0. \tag{36}$$

Note that in the case where $T$ is unitary, then $X$ is symplectic and orthogonal, and thus $Y = 0_{2M}$ recovering the results from the previous subsection.

# 3 One recurrence relation to rule them all

We can write $M$-mode pure states, mixed states, unitaries, and channels in the Fock space representation as

$$|\psi\rangle = \sum_{\boldsymbol{k}} \psi_{\boldsymbol{k}} |\boldsymbol{k}\rangle, \tag{37}$$

$$\rho = \sum_{\boldsymbol{j},\boldsymbol{k}} \rho_{\boldsymbol{j},\boldsymbol{k}} |\boldsymbol{j}\rangle\langle\boldsymbol{k}|, \tag{38}$$

$$U = \sum_{\boldsymbol{j},\boldsymbol{k}} U_{\boldsymbol{j},\boldsymbol{k}} |\boldsymbol{j}\rangle\langle\boldsymbol{k}|, \tag{39}$$

$$\Phi[|\boldsymbol{j}\rangle\langle\boldsymbol{l}|] = \sum_{\boldsymbol{i},\boldsymbol{k}} \Phi_{\boldsymbol{k},\boldsymbol{l},\boldsymbol{i},\boldsymbol{j}} |\boldsymbol{i}\rangle\langle\boldsymbol{k}|, \tag{40}$$

where the Fock space indices are expressed as a multi-index $\boldsymbol{k} = (k_1, k_2, \ldots, k_M)$. We now simplify the notation by considering the collections of amplitudes $\psi_{\boldsymbol{k}}$, $\rho_{\boldsymbol{j},\boldsymbol{k}}$, $U_{\boldsymbol{j},\boldsymbol{k}}$ and $\Phi_{\boldsymbol{i},\boldsymbol{j},\boldsymbol{k},\boldsymbol{l}}$ as instances of a tensor $\mathcal{G}_{\boldsymbol{k}}$ where $\boldsymbol{k}$ is $M$-dimensional for pure states, $2M$-dimensional for mixed states and unitary transformations, and $4M$-dimensional for channels.

One way to produce the Fock space amplitudes of a Gaussian object is to start from a generating function $\Gamma(\boldsymbol{\alpha})$ and then compute its derivatives. The generating function $\Gamma(\boldsymbol{\alpha})$ is also known as the stellar function [40] or the Bargmann function [21]. To obtain the generating function, one needs to contract each index of a Gaussian object with a *rescaled* multi-mode coherent state

$$e^{\frac{1}{2}\|\boldsymbol{\alpha}\|^2} |\boldsymbol{\alpha}\rangle, \tag{41}$$

where $\|.\|$ denotes the vector 2-norm. For example, for a pure state, we have

$$\Gamma_\psi(\boldsymbol{\alpha}) = e^{\frac{1}{2}\|\boldsymbol{\alpha}\|^2} \sum_{\boldsymbol{k}} \psi_{\boldsymbol{k}} \langle \boldsymbol{\alpha}^* | \boldsymbol{k}\rangle = \sum_{\boldsymbol{k}} \psi_{\boldsymbol{k}} \frac{\boldsymbol{\alpha}^{\boldsymbol{k}}}{\sqrt{\boldsymbol{k}!}} \tag{42}$$

$$= c_\psi \exp\left( \boldsymbol{\alpha}^T \boldsymbol{b}_\psi + \frac{1}{2} \boldsymbol{\alpha}^T A_\psi \boldsymbol{\alpha} \right), \tag{43}$$

where $A_\psi$ is an $M \times M$ complex symmetric matrix, $\boldsymbol{b}_\psi$ is an $M$-dimensional complex vector and $c_\psi$ is the vacuum amplitude.

In the case of density matrices, we obtain an analogous exponential as in (42), except that $A_\rho$ and $\boldsymbol{b}_\rho$ are of size $2M \times 2M$ and $2M$ respectively. For unitaries, $A_U$ and $\boldsymbol{b}_U$ are of size $2M \times 2M$ and $2M$, and for channels $A_\Phi$ and $\boldsymbol{b}_\Phi$ are of size $4M \times 4M$ and $4M$, respectively. Therefore, all Gaussian objects are characterized by a complex symmetric matrix $A$, a complex vector $\boldsymbol{b}$ and a complex scalar $c = \mathcal{G}_{\boldsymbol{0}}$, or conversely given valid $A$ and $\boldsymbol{b}$ and $c$ we can calculate the coefficients $\mathcal{G}_{\boldsymbol{k}}$ by computing derivatives of the appropriate order of the generating function $\Gamma(\boldsymbol{\alpha})$:

$$\mathcal{G}_{\boldsymbol{k}} = \mathcal{G}_{\boldsymbol{0}} \frac{\partial_{\boldsymbol{\alpha}}^{\boldsymbol{k}}}{\sqrt{\boldsymbol{k}!}} \exp\left( \boldsymbol{\alpha}^T \boldsymbol{b} + \frac{1}{2} \boldsymbol{\alpha}^T A \boldsymbol{\alpha} \right) \Bigg|_{\boldsymbol{\alpha}=\boldsymbol{0}}. \tag{44}$$

In this way we unify the calculation of the Fock space amplitudes of Gaussian objects into a single method that works in all cases, depending on which triple $(A, \boldsymbol{b}, c)$ one is considering.

In practice (as we will do in the following sections), it is sufficient to apply this method to the case of mixed states only, as the expressions split naturally thanks to the properties of the Hermite polynomials (see Eq. (61) to (65)), and one obtains the case of Gaussian pure states. For transformations, using the Choi-Jamiołkowski duality [41, 42] can treat channels as mixed states, and if a channel is unitary, the expressions split in the same way as they do for states (see Eq. (93) to (97)), and one obtains the case of Gaussian unitaries already treated in Ref. [35].

Multivariate derivatives of the exponential of a function can be computed with a linear recurrence formula [35], and in the case the function is a polynomial of degree $D$, the recurrence relation has order $D$. In our case, the polynomial has degree 2, which means we can write a linear recurrence relation of order 2 between the Fock space amplitudes:

$$\mathcal{G}_{\boldsymbol{k}+1_i} = \frac{1}{\sqrt{k_i+1}}\left(b_i \mathcal{G}_{\boldsymbol{k}} + \sum_j \sqrt{k_j} A_{ij} \mathcal{G}_{\boldsymbol{k}-1_j}\right), \tag{45}$$

with the vacuum amplitude initialized as $\mathcal{G}_{\boldsymbol{0}} = c$. In this recurrence relation, $\boldsymbol{k}+1_i$ is the vector $\boldsymbol{k}$ with the $i$-th element increased by 1 (and similarly for $\boldsymbol{k}-1_j$, where it is decreased by 1). We refer to $w = \sum_i k_i$ as the weight of the index. In essence, the recurrence relation allows us to write amplitudes of weight $w+1$ as linear combinations of amplitudes of weight $w$ and $w-1$. By applying it repeatedly, one can reach any Fock space amplitude (in practice, one eventually reaches a numerical precision horizon [43]).

## 3.1 Multidimensional Hermite polynomials

For reference, we recall the definition of the multidimensional Hermite polynomials as the Taylor series of a multidimensional Gaussian function, which has an additional factor of $\frac{1}{\sqrt{\boldsymbol{k}!}}$ with respect to the Fock amplitudes:

$$K^A(\boldsymbol{y}, \boldsymbol{b}) = \exp\left(\boldsymbol{y}^T \boldsymbol{b} + \tfrac{1}{2}\boldsymbol{y}^T A \boldsymbol{y}\right) = \sum_{\boldsymbol{k}\geq 0} \frac{G_{\boldsymbol{k}}^A(\boldsymbol{b})}{\boldsymbol{k}!} \boldsymbol{y}^{\boldsymbol{k}}. \tag{46}$$

Note the sign of the quadratic term in the exponential, which can differ from other conventions. In the last equation $\boldsymbol{b} \in \mathbb{C}^\ell$ is a complex vector, $A = A^T \in \mathbb{C}^{\ell\times\ell}$ is a complex symmetric matrix and $\boldsymbol{k} \in \mathbb{Z}_0^\ell$ is a vector of non-negative integers. This notation makes it explicit that

$$\left[\prod_{i=1}^\ell \left(\frac{\partial}{\partial y_i}\right)^{k_i}\right] K^A(\boldsymbol{y}, \boldsymbol{b})\Bigg|_{\boldsymbol{y}=0} = G_{\boldsymbol{k}}^A(\boldsymbol{b}). \tag{47}$$

These polynomials satisfy the recurrence relation

$$G_{\boldsymbol{k}+1_i}^A(\boldsymbol{b}) = b_i G_{\boldsymbol{k}}^A(\boldsymbol{b}) + \sum_{j=1}^M k_j A_{i,j} G_{\boldsymbol{k}-1_j}^A(\boldsymbol{b}), \tag{48}$$

where $1_i$ is a vector that has a 1 in the $i$-th entry and 0s elsewhere. Note that $G_{\boldsymbol{0}}^A(\boldsymbol{b}) = 1$, $G_{1_i}^A(\boldsymbol{b}) = b_i$ and that $G_{1_i+1_j}^A(\boldsymbol{b}) = b_i b_j + A_{ij}$. The multidimensional Hermite polynomial is related to the loop-hafnian function introduced in Ref. [44] which counts the number of perfect matchings of weighted graphs, including self-loops. They are related as follows

$$G_{\boldsymbol{k}}^A(\boldsymbol{b}) = \text{lhaf}(\text{fdiag}(A_{\boldsymbol{k}}, \boldsymbol{b}_{\boldsymbol{k}})), \tag{49}$$

where fdiag fills the diagonal of the matrix in the first argument using the vector in the second argument. Note that $A_{\boldsymbol{k}}$ is the matrix obtained from $A$ by repeating its $i$-th row and column $k_i$ times. Similarly, $\boldsymbol{b}_{\boldsymbol{k}}$ is the vector obtained from $\boldsymbol{b}$ by repeating its $i$-th entry $k_i$ times. Note that when $k_i = 0$ the relevant row and column of $A$ and entry of $\boldsymbol{b}$ are deleted. The best known methods to calculate the single loop-hafnian in Eq. (49) requires $O(C^3 \sqrt{\prod_{i=1}^\ell (1+k_i)})$ steps where $C$ is the number of nonzero entries in the vector $\boldsymbol{k}$ [45].

We will show below that the Fock representation of a pure Gaussian state, a mixed Gaussian state, a Gaussian unitary, or a Gaussian channel can all be written as

$$c \times \frac{G_{\mathbf{k}}^{A}(\mathbf{b})}{\sqrt{\mathbf{k}!}}, \tag{50}$$

where $c$ is a scalar, $\mathbf{b}$ is a vector of dimension $\ell$, $A$ is a square matrix of size $\ell \times \ell$ and $\mathbf{k} \in \mathbb{Z}_{\geq 0}^{\ell}$. The integer $\ell$ equals $M, 2M, 2M, 4M$ for pure states, mixed states, unitaries or channels on $M$ modes respectively.

Note that the quantity in Eq. (50) is potentially the ratio of two large numbers. In particular, since this quantity represents a probability or a probability amplitude it should be bounded in absolute value by 1. Thus it is often convenient, especially for numerical purposes, to introduce renormalized multidimensional Hermite polynomials as

$$\mathcal{G}_{\mathbf{k}}^{A}(\mathbf{b}) = c \times \frac{G_{\mathbf{k}}^{A}(\mathbf{b})}{\sqrt{\mathbf{k}!}}, \tag{51}$$

which satisfy the recurrence relation in Eq. (45).

Using results from Ref. [22] we can also find the differential of the matrix elements:

$$d\mathcal{G}_{\mathbf{k}}^{A}(\mathbf{b}) = \frac{[dc]}{c} \mathcal{G}_{\mathbf{k}}^{A}(\mathbf{b}) + \sum_{i=1}^{\ell} [db_i] \sqrt{k_i} \mathcal{G}_{\mathbf{k}-1_i}^{A}(\mathbf{b}) \tag{52}$$

$$+ \frac{1}{2} \sum_{i,j=1}^{\ell} [dA_{i,j}] \sqrt{k_i(k_j - \delta_{ij})} \mathcal{G}_{\mathbf{k}-1_i-1_j}^{A}(\mathbf{b}).$$

We can use this relation to write a new differential formula for the loop-hafnian with arbitrary repetitions that generalize the results in Ref. [46]

$$d\left[\mathrm{lhaf}(\mathrm{fdiag}(A_{\mathbf{k}}, \mathbf{b}_{\mathbf{k}}))\right] = \sum_{i=1}^{\ell} [db_i] k_i \mathrm{lhaf}(\mathrm{fdiag}(A_{\mathbf{k}-1_i}, \mathbf{b}_{\mathbf{k}-1_i})) \tag{53}$$

$$+ \frac{1}{2} \sum_{i,j=1}^{\ell} [dA_{i,j}] k_i(k_i - \delta_{ij}) \mathrm{lhaf}(\mathrm{fdiag}(A_{\mathbf{k}-1_i-1_j}, \mathbf{b}_{\mathbf{k}-1_i-1_j})).$$

Note that in the limit of no loops $\mathbf{b} = \mathbf{0}$ and no repetitions $k_i \in \{0, 1\}$ the last equation reproduces precisely Eq. (A12) of Ref. [46]. We note that one can obtain significant savings in traversing the recursions relations of these quantities by carefully exploiting symmetries [47].

## 3.2 States

In this subsection, we show how to turn the symplectic representation of a Gaussian state into the metaplectic or Fock space representation of the same object [21]. This follows the developments in Refs. [10–13, 32, 33].

To compute the Fock space amplitudes of a Gaussian pure state we need the triple $(A_\psi, \mathbf{b}_\psi, c_\psi)$ where $A_\psi$ and $\mathbf{b}_\psi$ are $M$-dimensional. If the state is mixed, we need the triple $(A_\rho, \mathbf{b}_\rho, c_\rho)$ where $A_\rho$ and $\mathbf{b}_\rho$ are $2M$-dimensional. We are now going to show how to obtain these triples.

It is convenient to introduce the $s-$parametrized complex covariance matrix

$$\boldsymbol{\sigma}_s = \boldsymbol{\sigma} + \frac{s}{2} \mathbb{1}_{2M}, \tag{54}$$

by definition $\boldsymbol{\sigma}_0 \equiv \boldsymbol{\sigma}$ and moreover we use the shorthand notation $\boldsymbol{\sigma}_\pm \equiv \boldsymbol{\sigma}_{\pm 1}$.

We recall the results derived in Ref. [12]. An expression for the metaplectic representation of the Gaussian state is

$$\langle \boldsymbol{m}|\rho|\boldsymbol{n}\rangle = c_\rho \times \prod_{s=1}^{M} \frac{\partial_{\alpha_s}^{n_s} \partial_{\alpha_s^*}^{m_s}}{\sqrt{n_s! m_s!}} \exp\left[\tfrac{1}{2} \boldsymbol{y}^T A_\rho \boldsymbol{y} + \boldsymbol{y}^T \boldsymbol{b}_\rho\right], \tag{55}$$

where, relative to Eq. (47), we identified $\boldsymbol{y} = \left[\begin{smallmatrix} \alpha \\ \alpha^* \end{smallmatrix}\right]$, $\boldsymbol{k} = \boldsymbol{n} \oplus \boldsymbol{m}$, $\ell = 2M$ and used the results from Refs. [10, 33, 48] to write together with the definitions in Eqs. (13) and (14)

$$A_\rho = P_M\left[\mathbb{1}_{2M} - \boldsymbol{\sigma}_+^{-1}\right] = P_M \boldsymbol{\sigma}_- \boldsymbol{\sigma}_+^{-1} = P_M \boldsymbol{\sigma}_+^{-1} \boldsymbol{\sigma}_-, \tag{56}$$

$$\boldsymbol{b}_\rho = \left(\boldsymbol{\sigma}_+^{-1} \bar{\boldsymbol{\mu}}\right)^* = P_M \boldsymbol{\sigma}_+^{-1} \bar{\boldsymbol{\mu}}, \tag{57}$$

$$c_\rho = \langle \boldsymbol{0}|\rho|\boldsymbol{0}\rangle = \frac{\exp\left[-\tfrac{1}{2} \bar{\boldsymbol{\mu}}^\dagger \boldsymbol{\sigma}_+^{-1} \bar{\boldsymbol{\mu}}\right]}{\sqrt{\det(\boldsymbol{\sigma}_+)}}, \tag{58}$$

$$P_M = \left[\begin{smallmatrix} 0_M & \mathbb{1}_M \\ \mathbb{1}_M & 0_M \end{smallmatrix}\right], \tag{59}$$

to finally write

$$\langle \boldsymbol{m}|\rho|\boldsymbol{n}\rangle = c_\rho \times \frac{G_{\boldsymbol{n}\oplus\boldsymbol{m}}^{A_\rho}(\boldsymbol{b}_\rho)}{\sqrt{\boldsymbol{n}! \boldsymbol{m}!}}. \tag{60}$$

The map $\boldsymbol{\sigma} \mapsto \boldsymbol{\sigma}_+^{-1} \boldsymbol{\sigma}_-$ in Eq. (56) is the Cayley transform [49,50]. In the case where $\rho = |\Psi\rangle\langle\Psi|$ is a pure state it is easy to show that

$$A_\rho = A_\psi^* \oplus A_\psi, \tag{61}$$

$$\boldsymbol{b}_\rho = \boldsymbol{b}_\psi^* \oplus \boldsymbol{b}_\psi, \tag{62}$$

and then

$$G_{\boldsymbol{n}\oplus\boldsymbol{m}}^{A_\rho}(\boldsymbol{b}_\rho) = G_{\boldsymbol{n}\oplus\boldsymbol{m}}^{A_\psi^* \oplus A_\psi}(\boldsymbol{b}_\psi^* \oplus \boldsymbol{b}_\psi) \tag{63}$$

$$= G_{\boldsymbol{n}}^{A_\psi^*}(\boldsymbol{b}_\psi^*) \times G_{\boldsymbol{m}}^{A_\psi}(\boldsymbol{b}_\psi) \tag{64}$$

$$= [G_{\boldsymbol{n}}^{A_\psi}(\boldsymbol{b}_\psi)]^* \times G_{\boldsymbol{m}}^{A_\psi}(\boldsymbol{b}_\psi), \tag{65}$$

which allows us to write the probability amplitude of a pure state

$$\langle \boldsymbol{m}|\Psi\rangle = c_\psi \frac{G_{\boldsymbol{m}}^{A_\psi}(\boldsymbol{b}_\psi)}{\sqrt{\boldsymbol{m}!}}, \quad c_\psi = e^{i\varphi_\Psi} \sqrt{c_\rho}, \tag{66}$$

up to a global phase $\varphi_\Psi$ that cannot be determined from the covariance matrix and vector of means of the pure Gaussian state. This will be discussed in a later section. Note that the last equation can be used to write the Hilbert-space ket representing the state as [11]

$$|\Psi\rangle = c_\psi \exp\left[\sum_{i=1}^{M} (b_\psi)_i a_i^\dagger + \tfrac{1}{2} \sum_{i,j=1}^{M} (A_\psi)_{i,j} a_i^\dagger a_j^\dagger\right] |\boldsymbol{0}\rangle, \tag{67}$$

thus showing that this formalism reduces to the one introduced by Krenn et al. in Refs. [51–54] when the displacements are zero. Moreover, the Gaussian formulation allows us to easily include the most common form of decoherence for bosonic modes, namely loss, since this process is a Gaussian channel.

We now give a few examples. A displaced squeezed state $\mathcal{D}(\alpha)\mathcal{S}(re^{i\phi})|0\rangle$ (which is the most general pure single-mode Gaussian state) has

$$\boldsymbol{A}_\psi = -\tanh(r)e^{i\phi}\,, \tag{68}$$

$$\boldsymbol{b}_\psi = \alpha + \alpha^* e^{i\phi}\tanh r\,, \tag{69}$$

$$c_\psi = \frac{\exp\left(-\frac{1}{2}\left[|\alpha|^2 + \alpha^{*2}e^{i\phi}\tanh r\right]\right)}{\sqrt{\cosh r}}\,, \tag{70}$$

and its amplitudes in the Fock basis satisfy

$$\psi_{k+1}^{\mathrm{dsq}} = \frac{1}{\sqrt{k+1}}\Big([\alpha + \alpha^* e^{i\phi}\tanh r]\psi_k^{\mathrm{dsq}} - \sqrt{k}\tanh(r)e^{i\phi}\psi_{k-1}^{\mathrm{dsq}}\Big). \tag{71}$$

In the limit of no squeezing, $r \to 0$ we obtain coherent states with recursion relation

$$\psi_{k+1}^{\mathrm{coh}} = \frac{1}{\sqrt{k+1}}\alpha\,\psi_k^{\mathrm{coh}}\,. \tag{72}$$

Similarly, in the limit of no displacement, $\alpha \to 0$ we obtain the recursion relation for squeezed vacuum states

$$\psi_{k+1}^{\mathrm{sq}} = -\sqrt{\frac{k}{k+1}}\tanh(r)e^{i\phi}\psi_{k-1}^{\mathrm{sq}}\,. \tag{73}$$

Note that, as expected, this recurrence relation skips odd indices.

For the simple case of $M$-mode squeezed states with real squeezing parameters $r_i$ sent into an interferometer with unitary $\boldsymbol{U}$ we have that $\boldsymbol{A}_\psi = -\boldsymbol{U}\left[\bigoplus_{i=1}^M \tanh r_i\right]\boldsymbol{U}^T$.

The thermal state is given by $\boldsymbol{A}_\rho = \frac{\bar{n}}{\bar{n}+1}\left(\begin{smallmatrix} 0 & 1 \\ 1 & 0 \end{smallmatrix}\right)$, $\boldsymbol{b}_\rho = \boldsymbol{0}$ and $c_\rho = \frac{1}{1+\bar{n}}$, where $\bar{n}$ is the average photon number, giving rise to the recurrence relations:

$$\rho_{k_1+1,k_2}^{\mathrm{th}} = \sqrt{\frac{k_2}{k_1+1}}\frac{\bar{n}}{\bar{n}+1}\rho_{k_1,k_2-1}^{\mathrm{th}}\,, \tag{74}$$

$$\rho_{k_1,k_2+1}^{\mathrm{th}} = \sqrt{\frac{k_1}{k_2+1}}\frac{\bar{n}}{\bar{n}+1}\rho_{k_1-1,k_2}^{\mathrm{th}}\,. \tag{75}$$

For a squeezed state along the $q$-quadrature with $r > 0$ (the symplectic matrix $\boldsymbol{S}$ can be found in Eq. (24)) that undergoes loss by transmission $\eta$ (defined in Eq. (32)), we start from the vacuum state with $\boldsymbol{V} = \frac{\hbar}{2}\mathbb{1}$, we apply the squeezing operator $\boldsymbol{V}' = \boldsymbol{S}\boldsymbol{V}\boldsymbol{S}^T$, we make the state pass through the lossy channel $\boldsymbol{V}'' = \boldsymbol{X}\boldsymbol{V}'\boldsymbol{X}^T + \boldsymbol{Y}$, and we obtain its covariance matrix $\boldsymbol{\sigma} = \frac{1}{\hbar}\boldsymbol{W}^\dagger\boldsymbol{V}''\boldsymbol{W}$. Then it is easy to find $\boldsymbol{A}_\rho$ from Eq. (56) that

$$\boldsymbol{A}_\rho = \frac{\eta}{\coth^2 r - (\eta-1)^2}\begin{bmatrix} -\coth r & 1-\eta \\ 1-\eta & -\coth r \end{bmatrix}\,. \tag{76}$$

In the limit of no loss we find $\boldsymbol{A}_\rho = -[\tanh r \oplus \tanh r]$ while in the limit of zero transmission we retrieve the single-mode vacuum, $\boldsymbol{A}_\rho = 0_2$.

## 3.3 Transformations

We can lift the description of states in the previous section to describe transformations via the Choi-Jamiołkowski duality in phase space, which allows us to faithfully map a channel by applying it over one-half of a full-rank entangled state. A Gaussian channel $\Phi[\cdot]$ is uniquely

determined by the triple $X, Y, d$ and acts on a Gaussian state as $(V, \bar{r}) \mapsto (XVX^T + Y, X\bar{r} + d)$. We can then write (see Appendix B and Appendix C for details)

$$\langle i | (\Phi[|j\rangle\langle l|]) | k \rangle = c_\Phi \times \frac{G^{A_\Phi}_{k \oplus l \oplus i \oplus j}(b_\Phi)}{\sqrt{i!j!k!l!}} , \tag{77}$$

where

$$A_\Phi = P_{2M} R \begin{bmatrix} \mathbb{1}_{2M} - \xi^{-1} & \xi^{-1} X \\ X^T \xi^{-1} & \mathbb{1}_{2M} - X^T \xi^{-1} X \end{bmatrix} R^\dagger , \tag{78}$$

$$b_\Phi = \frac{1}{\sqrt{\hbar}} R^* \begin{bmatrix} \xi^{-1} d \\ -X^T \xi^{-1} d \end{bmatrix} , \tag{79}$$

$$c_\Phi = \frac{\exp\left[-\frac{1}{2\hbar} d^T \xi^{-1} d\right]}{\sqrt{\det(\xi)}} , \tag{80}$$

and

$$R = \frac{1}{\sqrt{2}} \begin{bmatrix} \mathbb{1}_M & i\mathbb{1}_M & 0_M & 0_M \\ 0_M & 0 & \mathbb{1}_M & -i\mathbb{1}_M \\ \mathbb{1}_M & -i\mathbb{1}_M & 0_M & 0_M \\ 0_M & 0_M & \mathbb{1}_M & i\mathbb{1}_M \end{bmatrix} , \tag{81}$$

$$\xi = \frac{1}{2} \left( \mathbb{1}_{2M} + XX^T + \frac{2Y}{\hbar} \right) . \tag{82}$$

For example, for a single-mode amplifier channel with gain $g \geq 1$, we find

$$A_\Phi = \begin{bmatrix} 0 & \frac{1}{\sqrt{g}} & \frac{g-1}{g} & 0 \\ \frac{1}{\sqrt{g}} & 0 & 0 & 0 \\ \frac{g-1}{g} & 0 & 0 & \frac{1}{\sqrt{g}} \\ 0 & 0 & \frac{1}{\sqrt{g}} & 0 \end{bmatrix} , \qquad b_\Phi = 0, \qquad c_\Phi = 1/g . \tag{83}$$

For the case of the $M$-mode lossy interferometer with transmission matrix $T$ we find

$$A_\Phi = \begin{bmatrix} 0_M & T^* & 0_M & 0_M \\ T^\dagger & 0_M & 0_M & \mathbb{1}_M - T^\dagger T \\ 0_M & 0_M & 0_M & T \\ 0_M & \mathbb{1}_M - T^T T^* & T^T & 0_M \end{bmatrix} , \tag{84}$$

$$b_\Phi = 0, \tag{85}$$

$$c_\Phi = 1. \tag{86}$$

This identity allows us to find the probability of measuring an outcome photon number pattern $j = (j_1, \ldots, j_M)$ when the multimode Fock state $|i\rangle = |i_1\rangle \otimes \ldots \otimes |i_M\rangle$ is sent into a lossy interferometer with transmission matrix $T$ (cf. Appendix E)

$$\langle j | \Phi_T[|i\rangle\langle i|] | j \rangle = \frac{1}{i!j!} \text{perm}\left( \begin{bmatrix} \mathbb{1}_M - T^\dagger T & T^\dagger \\ T & 0 \end{bmatrix}_{j \oplus i} \right) , \tag{87}$$

where perm is the permanent. The last equation reduces to the well-known lossless [55–57] case when $\mathbb{1}_M - T^\dagger T = 0$. Finally, note that we can obtain the single-mode pure loss channel by energy transmission $\eta$ by setting $T = \sqrt{\eta}$ in Eq. (84) to obtain

$$A_\Phi = \begin{bmatrix} 0 & \sqrt{\eta} & 0 & 0 \\ \sqrt{\eta} & 0 & 0 & 1-\eta \\ 0 & 0 & 0 & \sqrt{\eta} \\ 0 & 1-\eta & \sqrt{\eta} & 0 \end{bmatrix} . \tag{88}$$

This substitution illustrates an elegant property of our formalism, namely that Gaussian dual channels are related to each other by permuting even and odd blocks of rows and columns as can be seen by comparing Eq. (88) and Eq. (83) showing that indeed pure loss and amplification are duals of each other.

Our formalism can also handle non-trace-preserving maps. For example we can express the Fock damping channel as

$$\Phi[\rho] = e^{-\beta a^\dagger a} \rho e^{-\beta a^\dagger a}, \tag{89}$$

which has

$$A_\Phi = e^{-\beta} \begin{bmatrix} 0 & 1 & 0 & 0 \\ 1 & 0 & 0 & 0 \\ 0 & 0 & 0 & 1 \\ 0 & 0 & 1 & 0 \end{bmatrix}, \qquad b_\Phi = 0, \qquad c_\Phi = 1. \tag{90}$$

Note that the Fock damping operator $e^{-\beta a^\dagger a}$ itself has triple:

$$A = e^{-\beta} \begin{bmatrix} 0 & 1 \\ 1 & 0 \end{bmatrix}, \qquad b = 0, \qquad c = 1. \tag{91}$$

In the case where the channel is unitary, we can write $\Phi[\cdot] = U\{\cdot\}U^\dagger$ and then we obtain

$$\langle i | (\Phi[|j\rangle\langle l|]) | k \rangle = \langle i | U | j \rangle \langle l | U^\dagger | k \rangle. \tag{92}$$

This corresponds to the case where $Y = 0_{2M}$ and $X = S$ is symplectic. As we show in the Appendix D, we can then write

$$A_\Phi = A_U^* \oplus A_U, \tag{93}$$

$$b_\Phi = b_U^* \oplus b_U, \tag{94}$$

and then we have

$$\langle i | (\Phi[|j\rangle\langle l|]) | k \rangle = \frac{G^{A_U^* \oplus A_U}_{k\oplus l \oplus i \oplus j}(b_U^* \oplus b_U)}{\sqrt{i!j!k!l!}} \tag{95}$$

$$= \frac{G^{A_U^*}_{k\oplus l}(b_U^*)}{\sqrt{k!l!}} \times \frac{G^{A_U}_{i\oplus j}(b_U)}{\sqrt{i!j!}} \tag{96}$$

$$= \frac{\left[G^{A_U}_{k\oplus l}(b_U)\right]^*}{\sqrt{k!l!}} \times \frac{G^{A_U}_{i\oplus j}(b_U)}{\sqrt{i!j!}}. \tag{97}$$

Comparing Eq. (92) and the last equation we easily identify

$$\langle i | U | j \rangle = c_U \frac{G^{A_U}_{i\oplus j}(b_U)}{\sqrt{i!j!}}, \qquad c_U = \sqrt{c_\Phi} e^{i\varphi_U}, \tag{98}$$

where $\varphi_U$ is a phase that will be discussed in the next section. Note that the quantities $c_U, b_U$ and $A_U$ correspond to the $C, \mu, -\Sigma$ introduced in Eq. (26) of Ref. [22]. This comparison also allows us to conclude that $A_U$ is not only symmetric but also unitary (this can also be seen by inspecting the form of $A_U$ in Eq. (D.10) in the Appendix D).

Table 1: We derive the triple $A, b, c$ for the channel. We also generalize the results for transformations in Refs. [22,34] and those for pure states in Refs. [11,32] and mixed states in Refs. [10, 12, 33]. Moreover, we show the relation of the triple between channel and transformation, as well as between mixed state and pure state.

| Object | $A$ | $b$ | $c$ | Refs. |
|---|---|---|---|---|
| Channel $\Phi$ | $P_{2M}R\begin{bmatrix} \mathbb{1}_{2M}-\xi^{-1} & \xi^{-1}X \\ X^T\xi^{-1} & \mathbb{1}_{2M}-X^T\xi^{-1}X \end{bmatrix}R^\dagger$ | $\frac{1}{\sqrt{\hbar}}R^*\begin{bmatrix} \xi^{-1}d \\ -X^T\xi^{-1}d \end{bmatrix}$ | $\frac{\exp\left[-\frac{1}{2\hbar}d^T\xi^{-1}d\right]}{\sqrt{\det(\xi)}}$ | This work |
| Transformation $U$ | $A_\Phi = A_U^* \oplus A_U$ | $b_\Phi = b_U^* \oplus b_U$ | $c_\Phi = c_U^* c_U$ | [22,34] |
| Mixed state $\rho$ | $P_M(\mathbb{1}_{2M}-\sigma_+^{-1})$ | $P_M\sigma_+^{-1}\bar{\mu}$ | $\frac{\exp\left[-\frac{1}{2}\bar{\mu}^\dagger\sigma_+^{-1}\bar{\mu}\right]}{\sqrt{\det(\sigma_+)}}$ | [10,12,33] |
| Pure state $\psi$ | $A_\rho = A_\psi^* \oplus A_\psi$ | $b_\rho = b_\psi^* \oplus b_\psi$ | $c_\rho = c_\psi^* c_\psi$ | [11,32] |

## 4 Global phase of the Fock representation

In the Gaussian representation, transformations are specified by a symplectic matrix and a displacement vector. However, these two quantities do not uniquely specify the evolution of a quantum state. For example, when two displacement operators with parameters $d_1$ and $d_2$ are composed in the Gaussian representation, their effect is just another displacement with parameter $d = d_1 + d_2$. However, the unitary representation acquires a global phase:

$$\mathcal{D}(\alpha)\mathcal{D}(\beta) = e^{(\alpha\beta^* - \alpha^*\beta)/2}\mathcal{D}(\alpha+\beta), \tag{99}$$

i.e. we do not only add up both displacement parameters $\mathcal{D}(\alpha+\beta)$ here, but also get an extra part $e^{(\alpha\beta^* - \alpha^*\beta)/2}$, which is a *global phase*. Such a global phase is important when evolving linear combinations of Gaussian states with Gaussian operations [36]. This section will compute this global phase and provide some examples.

We know that the Fock representation of an arbitrary Gaussian unitary transformation is parametrized by the triple $(A_U, b_U, c_U)$. The unitary representation of the combination of two Gaussian transformations $\mathcal{U}_f = \mathcal{U}_1\mathcal{U}_2$ may have an additional global phase and we are going to find it.

We begin by calculating the Husimi $Q(\beta, \beta')$ function of the composition of $\mathcal{U}_1$ and $\mathcal{U}_2$ and we use a resolution of the identity in terms of coherent states to write:

$$\langle\beta^*|\mathcal{U}_1\mathcal{U}_2|\beta'\rangle = \langle\beta^*|\mathcal{U}_1 I \mathcal{U}_2|\beta'\rangle \tag{100}$$

$$= \frac{1}{\pi^M}\int_{\mathbb{C}^M} d^{2M}\alpha\,\langle\beta^*|\mathcal{U}_1|\alpha\rangle\langle\alpha|\mathcal{U}_2|\beta'\rangle, \tag{101}$$

where we can replace the Husimi $Q$ functions for generic Gaussian transformations $\langle\beta^*|\mathcal{U}_1|\alpha\rangle$ and $\langle\alpha|\mathcal{U}_2|\beta'\rangle$. After integrating $\alpha$, the $Q$ function of the composite operator $\mathcal{G}_f$ is obtained, which is characterized by:

$$A_{U_f} = B_1 \oplus D_2 + \left\{C_1 \oplus C_2^T\right\}\mathcal{Z}\left\{C_1^T \oplus C_2\right\}, \tag{102}$$

$$b_{U_f}^T = [c_1^T, d_2^T] + [d_1^T, c_2^T]\mathcal{Z}\left\{C_1^T \oplus C_2\right\}, \tag{103}$$

$$c_{U_f} = \frac{c_{U_1}c_{U_2}}{\sqrt{\det(\mathcal{Y})}}\exp\left(\frac{1}{2}[d_1^T, c_2^T]\mathcal{Z}\begin{bmatrix} d_1 \\ c_2 \end{bmatrix}\right), \tag{104}$$

where $\boldsymbol{b}_{U_i}^T$ and $\boldsymbol{A}_{U_i}$ are written in block form:

$$\boldsymbol{b}_{U_i}^T = \left[\boldsymbol{c}_i^T, \boldsymbol{d}_i^T\right], \tag{105}$$

$$\boldsymbol{A}_{U_i} = \left[\begin{array}{c|c} \boldsymbol{B}_i & \boldsymbol{C}_i \\ \hline \boldsymbol{C}_i^T & \boldsymbol{D}_i \end{array}\right], \tag{106}$$

and we introduce the auxiliary quantities:

$$\mathcal{Y} = \mathbb{1}_M - \boldsymbol{D}_1 \boldsymbol{B}_2, \tag{107}$$

$$\mathcal{Z} = \mathcal{Z}^T = \left[\begin{array}{cc} -\boldsymbol{D}_1 & \mathbb{1}_M \\ \mathbb{1}_M & -\boldsymbol{B}_2 \end{array}\right]^{-1} = \left[\begin{array}{cc} \mathcal{Y}^{-1}\boldsymbol{B}_2 & \mathcal{Y}^{-1} \\ [\mathcal{Y}^T]^{-1} & \boldsymbol{D}_1 \mathcal{Y}^{-1} \end{array}\right]. \tag{108}$$

Eq. (104) gives the global phase for the composite Gaussian operator. The details of this calculation can be found in Appendix F.

As examples, we show the composition of two single-mode displacements and the composition of two single-mode squeezers. Recall that they correspond to

$$\mathcal{D}(\alpha): \boldsymbol{A}_U = \begin{pmatrix} 0 & 1 \\ 1 & 0 \end{pmatrix}, \qquad \boldsymbol{b}_U = [\alpha, -\alpha^*]^T, \qquad c_U = e^{-\frac{1}{2}|\alpha|^2}, \tag{109}$$

$$\mathcal{S}(re^{i\delta}): \boldsymbol{A}_U = \begin{pmatrix} -e^{i\delta}\tanh r & \operatorname{sech} r \\ \operatorname{sech} r & e^{-i\delta}\tanh r \end{pmatrix}, \tag{110}$$

$$\boldsymbol{b}_U = [0,0], \qquad c_U = \frac{1}{\sqrt{\cosh r}}.$$

For a composition of displacement operators $\mathcal{D}(\alpha)\mathcal{D}(\beta)$, we have

$$\det(\mathbb{1}_M - \boldsymbol{D}_1 \boldsymbol{B}_2) = 1, \quad \mathcal{Y} = 1. \tag{111}$$

We then obtain the global phase:

$$c_{U_f} = c_{U_\alpha} c_{U_\beta} \exp\left(-\alpha^* \beta\right) = c_{U_{(\alpha+\beta)}} \exp\left(\tfrac{1}{2}\alpha\beta^* - \tfrac{1}{2}\beta\alpha^*\right), \tag{112}$$

recovering Eq. (99).

For two squeezers $\mathcal{S}(\zeta_1), \mathcal{S}(\zeta_2)$, since $\boldsymbol{b}_U$ is zero, we have

$$\det(\mathbb{1}_M - \boldsymbol{D}_1 \boldsymbol{B}_2) = 1 + e^{i(\delta_2 - \delta_1)}\tanh r_1 \tanh r_2, \tag{113}$$

and in turn, we get

$$c_{U_f} = \frac{c_{U_1} c_{U_2}}{\sqrt{\det(\mathbb{1}_M - \boldsymbol{D}_1 \boldsymbol{B}_2)}} \tag{114}$$

$$= \frac{\sqrt{\operatorname{sech} r_1 \operatorname{sech} r_2}}{\sqrt{1 + e^{i(\delta_2 - \delta_1)}\tanh r_1 \tanh r_2}}, \tag{115}$$

which coincides with the results from Refs [58, 59].

Finally, note that when composing two passive Gaussian unitaries we already know that there is no extra phase since by construction (cf. Eq. (25)) $\langle \boldsymbol{n}|\mathcal{W}(\boldsymbol{J})|\boldsymbol{0}\rangle = \delta_{\boldsymbol{n},\boldsymbol{0}}$.

# 5 Learning Gaussian states and transformations

Differentiability is a desirable property for a computational model, as it enables gradient descent optimization. Suppose one can write a cost function $L$ in terms of an independent variable (or collection of variables) $\theta$, then the idea of gradient descent is to update the independent variable by taking an optimization step on the opposite of the gradient $\partial L$ repeatedly thus converging to a local minimum of the cost function.

Normally, we optimize each fundamental Gaussian operator inside the circuit, such as the displacement, the squeezers and etc. However, with the increasing number of modes, we obtain a more complicated circuit and the update of each variable becomes heavy work. That is why we propose the idea to optimize a *single* Gaussian object (which can be decomposed into the fundamental Gaussian operators).

All Gaussian objects can be updated in a learning step on the symplectic group, on the displacement parameters, or on the symplectic eigenvalues. As the latter two are Euclidean updates, we will not describe them in great detail. In fact, once the relevant Euclidean gradient has been computed, the update rule can be taken as a single step of gradient descent or one of its variants (e.g. using momentum). For instance, the update of the displacement parameter could simply follow the rule

$$d \leftarrow d - t \frac{\partial L}{\partial d}, \tag{116}$$

using the Euclidean gradient and $t$ is the learning rate.

We will concentrate then on detailing the update on the symplectic group $S$, which is endowed in the Riemannian manifold. This section summarizes the basic ideas of gradient descent on Riemannian manifolds, particularly on the manifold of symplectic matrices and unitary matrices. In the end, we comment on this global Gaussian operator optimization idea.

Note that in the first four subsections below, the symbols $A$, $B$, $M$, $p$, $R$, $W$, $X$, $Y$, $Z$, $\gamma$ are defined locally and do not correspond to previous uses.

## 5.1 The symplectic group

We describe the manifold of real symplectic $2n \times 2n$ matrices as an embedded sub-manifold of $\mathbb{R}^{2n \times 2n}$:

$$\mathrm{Sp}(2n, \mathbb{R}) = \{S \in \mathbb{R}^{2n \times 2n} | S\Omega S^T = \Omega\}, \tag{117}$$

where $\Omega$ is defined in Eq.(8). Given that the condition $S\Omega S^T = \Omega$ is quadratic in $S$, the manifold of symplectic matrices is not a linear subspace of $\mathbb{R}^{2n \times 2n}$, which means that we likely *leave* the manifold after a naive straight step of gradient descent. In this section, we explain how to overcome this difficulty.

Note that unless details are relevant, we abbreviate $\mathrm{Sp}(2n, \mathbb{R})$ with Sp.

## 5.2 Tangent and normal spaces

If we differentiate the quadratic condition $S\Omega S^T = \Omega$ we obtain the *linear* tangency condition $X\Omega S^T + S\Omega X^T = 0$. All the matrices $X$ that satisfy the new condition form a linear subspace of $\mathbb{R}^{2n \times 2n}$ called the tangent space of Sp at the point $S$:

$$T_S \mathrm{Sp} = \{X \in \mathbb{R}^{2n \times 2n} | X\Omega S^T + S\Omega X^T = 0_{2n}\} \tag{118}$$

$$= \{S\Omega A | A = A^T\}. \tag{119}$$

Eq. (119) is a compact way of parametrizing the tangent space at $S$ using symmetric matrices. It can be found by imposing $X = S\Omega A$ in the tangency condition.

As a special case, the Lie algebra of Sp is the tangent space at the identity, i.e.

$$sp(2n, \mathbb{R}) = T_e Sp(2n, \mathbb{R}) \tag{120}$$

$$= \{X \in \mathbb{R}^{2n \times 2n} | X\Omega + \Omega X^T = 0_{2n}\} \tag{121}$$

$$= \{\Omega A | A = A^T\}. \tag{122}$$

We can then define the normal space at $S$ as the linear space containing all the elements that are orthogonal to $T_S Sp$:

$$N_S Sp = \{W \in \mathbb{R}^{2n \times 2n} | \text{Tr}(W^T X) = 0_{2n}, X \in T_S Sp\} \tag{123}$$

$$= \{\Omega S B | B = -B^T\}, \tag{124}$$

with Eq. (124) showing that we can parametrize the normal space at each point in Sp using anti-symmetric matrices.

## 5.3   Riemannian metric on Sp($2n$)

A Riemannian manifold such as Sp($2n, \mathbb{R}$) comes equipped with an inner product $\langle \cdot, \cdot \rangle_S$ on the tangent space $T_S Sp$ at each point $S \in Sp$. The family of inner products forms the Riemannian metric tensor. The inner product in $T_S Sp$ is defined as

$$\langle X, Y \rangle_S = \langle S^{-1} X, S^{-1} Y \rangle = \langle RX, Y \rangle, \tag{125}$$

where $R = S^{-T} S^{-1} = \Omega S S^T \Omega^T$ and note that $R^{-1} = SS^T$.

Consider now a cost function $L : Sp \to \mathbb{R}$. The Euclidean gradient $\partial L$ at the point $S$ (which is computed using the embedding coordinates in $\mathbb{R}^{2n \times 2n}$) is related to the Riemannian gradient $\nabla L \in T_S Sp$ by the compatibility condition

$$\langle \nabla L, X \rangle_S = \langle \partial L, X \rangle, \quad \forall X \in T_S Sp. \tag{126}$$

After rearranging the terms, the condition is equivalent to

$$\langle R\nabla L - \partial L, X \rangle = 0, \quad \forall X \in T_S Sp. \tag{127}$$

This means that $R\nabla L - \partial L \in N_S Sp$ and therefore, it must be possible to write

$$R\nabla L - \partial L = \Omega S B, \tag{128}$$

for some anti-symmetric matrix $B$. At the same time we have the tangency condition $\nabla L \Omega S^T + S\Omega \nabla^T L = 0$. If we replace $\nabla L$ from Eq. (128) into the tangency condition, we obtain an expression for $B$ and we can finally write the Riemannian gradient on the symplectic group:

$$\nabla L = \frac{S}{2}(Z + \Omega Z^T \Omega), \tag{129}$$

where $Z = S^T \partial L$.

The symplectic matrix that describes an interferometer belongs to the intersection of the orthogonal group O($2n$) and the symplectic group Sp($2n$), which is a unitary group U($n$):

$$U(n) = \{M \in \mathbb{C}^{n \times n} | M^\dagger M = MM^\dagger = \mathbb{1}_n\}. \tag{130}$$

We can go through the same arguments as with the symplectic group and obtain the Riemannian gradient in the unitary group (More calculation details are in Appendix G):

$$\nabla L = \frac{M}{2}(Z - Z^\dagger), \tag{131}$$

where $Z = M^\dagger \partial L$.

## 5.4 Geodesic optimization on Sp($2n$) and U($n$)

The shortest curve connecting two points on a Riemannian manifold $\mathcal{M}$ is called a geodesic, and it can be defined by the starting point $\gamma(0) = \boldsymbol{p}$ and its velocity on the tangent space at that point: $\boldsymbol{V} = \dot{\gamma}(0) \in T_{\boldsymbol{p}}\mathcal{M}$. For the symplectic and unitary groups, geodesics take the following form (which can be found by minimizing a variational formulation of the path length between two points [60,61]):

$$\gamma^{\text{Sp}(2n)}(t) = \boldsymbol{S} e^{t(\boldsymbol{S}^{-1}\boldsymbol{V})^T} e^{t[\boldsymbol{S}^{-1}\boldsymbol{V} - (\boldsymbol{S}^{-1}\boldsymbol{V})^T]}, \tag{132}$$

$$\gamma^{\text{U}(n)}e(t) = \boldsymbol{M} e^{t(\boldsymbol{M}^{\dagger}\boldsymbol{V})^T}. \tag{133}$$

By using a geodesic, we guarantee that each update step remains on the manifold.

For gradient descent, we use $\boldsymbol{V} = -\nabla L$:

$$\gamma^{\text{Sp}(2n)}(t) = \boldsymbol{S} e^{-t\boldsymbol{Y}} e^{-t(\boldsymbol{Y} - \boldsymbol{Y}^T)}, \tag{134}$$

with $\boldsymbol{Y} = \boldsymbol{S}^{-1}\nabla L = \frac{1}{2}(\boldsymbol{Z} + \boldsymbol{\Omega}\boldsymbol{Z}^T\boldsymbol{\Omega})$. For the unitary group, we obtain

$$\gamma^{\text{U}(n)}(t) = \boldsymbol{M} e^{-t\boldsymbol{Y}}, \tag{135}$$

with $\boldsymbol{Y} = \boldsymbol{M}^{\dagger}\nabla L = \frac{1}{2}(\boldsymbol{Z} - \boldsymbol{Z}^{\dagger}) = \frac{1}{2}(\boldsymbol{M}^{\dagger}\partial L - (\partial L)^{\dagger}\boldsymbol{M})$. We now have a geodesic update formula that we can apply in place of the usual gradient descent step. The parameter $t$ takes the role of the learning rate (which we fix depending on the application). For the symplectic group, we have

$$\boldsymbol{Z}_k \leftarrow \boldsymbol{S}_k^T \partial L, \tag{136}$$

$$\boldsymbol{Y}_k \leftarrow \frac{1}{2}(\boldsymbol{Z}_k + \boldsymbol{\Omega}\boldsymbol{Z}_k^T\boldsymbol{\Omega}), \tag{137}$$

$$\boldsymbol{S}_{k+1} \leftarrow \boldsymbol{S}_k e^{-t\boldsymbol{Y}_k} e^{-t(\boldsymbol{Y}_k - \boldsymbol{Y}_k^T)}. \tag{138}$$

For the unitary group, we have

$$\boldsymbol{Z}_k \leftarrow \boldsymbol{M}_k^{\dagger}\partial L, \tag{139}$$

$$\boldsymbol{Y}_k \leftarrow \frac{1}{2}(\boldsymbol{Z}_k - \boldsymbol{Z}_k^{\dagger}), \tag{140}$$

$$\boldsymbol{M}_{k+1} \leftarrow \boldsymbol{M}_k e^{-t\boldsymbol{Y}_k}. \tag{141}$$

Finally, we obtain the orthogonal matrix of the interferometer using Eq. (34).

## 5.5 The Riemannian update step in practice

We concentrate now on detailing the update on the symplectic group and we will take Gaussian unitaries as a basic example (pure states, mixed states, and channels can have a symplectic matrix among their parameters, via the Choi-Jamiołkowski duality).

The backpropagation procedure of the gradient calculation is shown in Fig. 1. The Euclidean gradient of the symplectic matrix can be calculated via the chain rule:

$$\frac{\partial L}{\partial \boldsymbol{S}} = 2\Re \left[ \sum_{X=\boldsymbol{A}, \boldsymbol{b}, c} \sum_k \frac{\partial L}{\partial \mathcal{G}_k} \frac{\partial \mathcal{G}_k}{\partial X} \frac{\partial X}{\partial \boldsymbol{S}} \right]. \tag{142}$$

In this expression, $\frac{\partial L}{\partial \mathcal{G}_k}$ is the upstream gradient which can be obtained from an Automatic Differentiation (AD) framework such as TensorFlow, $\frac{\partial \mathcal{G}_k}{\partial X}$ is computed by differentiating the

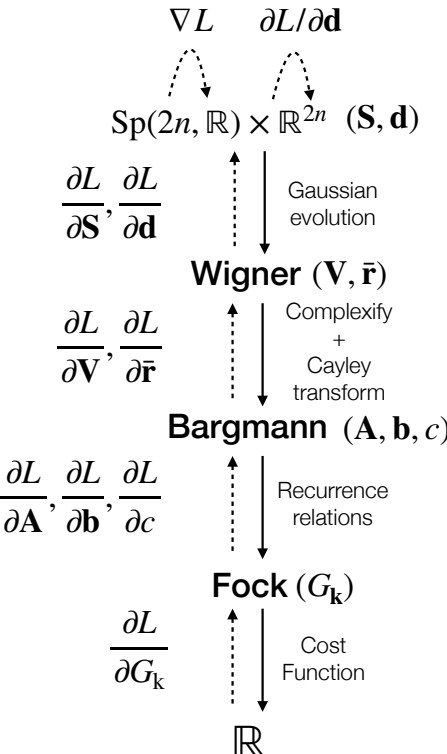

Figure 1: The detailed forward and backward passes. The Riemannian gradient $\nabla L$ for the geodesic update is calculated via the chain rule and Eq. (129), which backpropagates the gradient of the cost function with respect to the Fock amplitudes $\frac{\partial L}{\partial G_k}$ all the way to $\nabla L$, while the gradient $\frac{\partial L}{\partial d}$ is used directly to optimize $d$ on $\mathbb{R}^{2n}$. The backpropagation steps can be left to an Automatic Differentiation framework, except for the Fock to Bargmann step and the conversion between Euclidean and Riemannian gradient, which we implement ourselves.

recurrence relation in Eq. (45) and $\frac{\partial X}{\partial S}$ is also handled by the AD framework, and it depends on the functional relation between the symplectic matrix and $X$ denotes the triple $(A, b, c)$ we defined in section 3.

Then, we can write the update rule for the real symplectic matrix $S$ to follow a geodesic path starting at $S$ with a velocity $\nabla L$ defined by its Riemannian gradient and guarantee the updated matrix is still on $\mathrm{Sp}(2n)$.

## 5.6 Discussion

Our *single* Gaussian object optimization idea, using Riemannian gradient descent, has several advantages compared with the optimization of each circuit component separately, which we call Euclidean optimization.

Firstly, the optimization of the circuit before any decomposition in terms of gates with Euclidean parameters can be considered as the first advantage of our method, which can be useful for answering theoretical questions involving an extremization over the entire class of Gaussian states or transformations.

Secondly, our method gives more accurate results than optimizing each component separately in the Fock representation. This is because to obtain the Fock representation of the complete circuit one first needs to compute the Fock amplitudes of the circuit elements up to a cutoff for each of the elements and then multiply them all together. The inaccuracy of each element in the Fock amplitudes will come from the truncation of Fock space and will accumu-

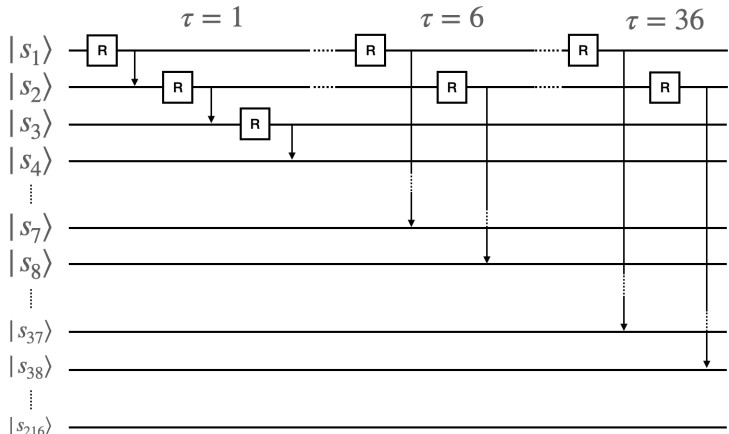

Figure 2: Schematic of the connectivity of the 216-mode circuit of the photonic processor Borealis [64], where all the nearest neighbour modes are connected by beam splitters, then all pairs at distance 6 and finally all pairs at distance 36. The arrows represent beam splitters.

late by contracting each of them. However, our Riemannian optimization keeps them together as a single Gaussian object (which is equivalent to contracting them in an infinite-dimensional Fock space).

Last but not least, the Riemannian optimization runs faster and can converge in fewer steps than the Euclidean optimization. In Appendix I, we show that by choosing the same learning rate of three different methods, the Riemannian optimization converges much faster than Euclidean optimization in the task of preparing a cat state. Also, some interesting optimization questions are raised along with the results.

# 6   Numerical experiments

In this section, we showcase the optimization methods introduced in the previous sections with three examples. The recurrent methods presented here are implemented in the open-source library TheWalrus [62] and they are integrated with the optimization methods in the open-source library MrMustard [1].

## 6.1   Minimizing the sparsity of the adjacency matrix of high-dimensional Gaussian boson sampling instance

We first analyze high-dimensional Gaussian Boson Sampling (GBS) [63] instances similar to the 216-mode circuit of the photonic processor Borealis [64]. This is made possible by working in phase space, as all the components are Gaussian and the cost function involves the $A$ matrix of the output state (i.e. not its Fock amplitudes). In a $D-$dimensional high-dimensional GBS instance with $M = d^D$ modes, a set of $K \leq M$ squeezed modes are sent into an interferometer composed of layers of beamsplitter gates (with a local rotation gate in the first mode) between modes $i$ and $i + \tau$ with $\tau \in \{1, d, d^2, \ldots, d^{D-1}\}$ as shown schematically for $d = 6$ and $D = 3$ in Fig. 2.

One desirable property of any GBS instance is that its adjacency matrix, which corresponds to $A_\psi$ in our notation, should not have any special property like being banded, sparse, or low-rank. This is because these types of properties can be exploited to speed-up the classical simulation of GBS.

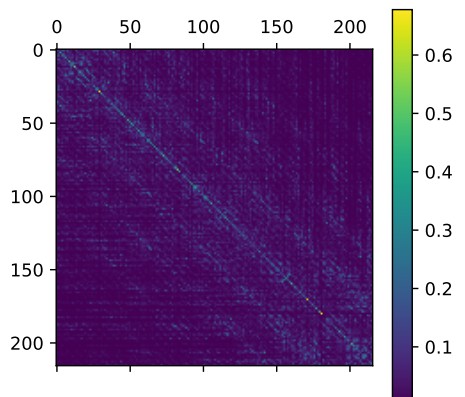

(a) The absolute value of the original matrix $A$.

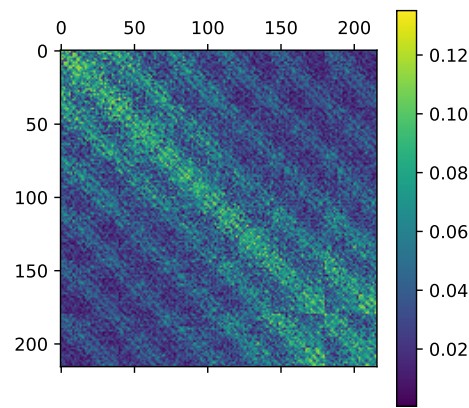

(b) The absolute value of the matrix $A$ after optimization.

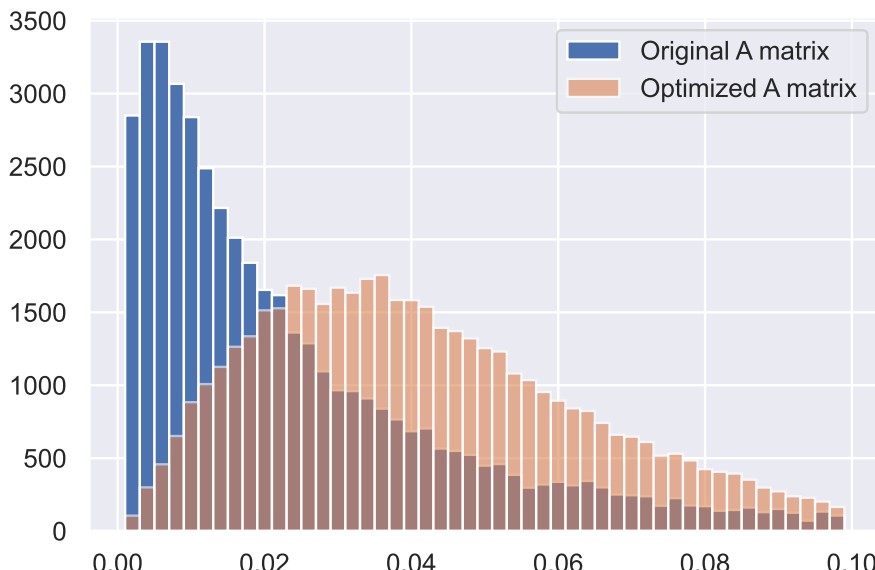

(c) Histogram of the absolute values of $A$ matrix elements before and after optimization.

Figure 3: Maximizing the entanglement in Borealis.

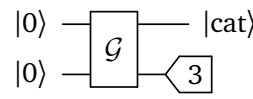

(a) Two-mode Gaussian transformation with PNR on the last mode.

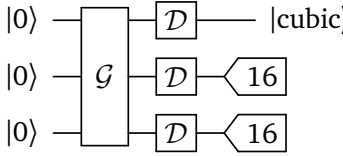

(b) Three-mode Gaussian transformation, followed by three displacements and photon number projections on the last two modes.

Figure 4: Circuits optimized in the examples.

For high-dimensional GBS instances like the one implemented in Borealis, it is known that the $A_\psi$ is full-rank (since every input is squeezed) and not banded (due to the long-ranged gates). However, one needs to judiciously choose the parameters of the beamsplitter so that the distribution of its entries is not heavily dominated by just a few of them. For example, if the one chooses the rotation gates and the transmission angles of the beamsplitters to be uniformly random in $[-\frac{\pi}{2}, \frac{\pi}{2}]$ one obtains the distribution shown in blue bars in Fig. 3c and the $A_\psi$ matrix show in Fig. 3a. For these results and following Ref. [64] we fix the phase angle of the beamsplitter to be $\pi/2$, we set the input squeezing parameter in all the modes to be $r = \operatorname{arcsinh} 1 \approx 0.8813736$ and take $D = 3$, $d = 6$ and thus a total of $M = 6^3 = 216$ modes. Note that the values of the matrix are heavily concentrated, i.e., for each row and column a few values are overwhelmingly larger than the rest.

We can now use the methods we developed to try to spread-out as much as possible the entries of the matrix $A_\psi$ thus we optimize the cost function

$$\min \sum_{ij} (|A_\psi|_{ij}^2 - \operatorname{mean}|A_\psi|^2)^2 \, . \tag{143}$$

We perform this optimization obtaining the distribution shown with the orange bars in Fig. 3c and the matrix shown in Fig. 3b. Notice that now the values are more evenly distributed.

## 6.2 State preparation

In this section, we find explicit circuits that prepare cat states and cubic phase states. For the cat state preparation we optimize a 2-mode Gaussian state with 3 photons measured in its last mode. For the cubic phase state preparation we optimize a 3-mode Gaussian state with 16 and 16 photons measured in its last two modes.

### 6.2.1 Cat state

The cat state that we target is the superposition of two coherent states:

$$|\mathrm{cat}_\pm\rangle = \frac{|\alpha\rangle \pm |-\alpha\rangle}{\sqrt{2 \pm 2e^{-2|\alpha|^2}}} \, , \tag{144}$$

where $|\alpha\rangle = \mathcal{D}(\alpha)|0\rangle$ is a coherent state. In the last equation the plus and minus signs corresponds to even and odd cat states, respectively.

For this example, we will target the generation of an odd cat state with $\alpha = 2$ and will employ the symplectic optimizer in MrMustard (version 0.5.0).

The first circuit (shown in Fig. 4a) consists of a Gaussian transformation followed by a measurement of 3 photons on the second mode and generates the (approximate) cat state in

the first mode. We use the symplectic optimizer to train the Gaussian gate. The result is shown in Fig. 5a with a fidelity of 99.37% and 7.47% success probability.

The code snippet below corresponds to the circuit shown in Fig. 4a:

```python
import numpy as np
from mrmustard.lab import *
from mrmustard.physics import fidelity, normalize
from mrmustard.training import Optimizer
from mrmustard import settings

alpha = 2.0   # coherent state amplitude
cutoff = 100  # fock space cutoff

cat_amps = (Coherent(alpha).ket([cutoff])
            - Coherent(-alpha).ket([cutoff]))
cat_target = normalize(State(ket=cat_amps))

# randomly initialized 2-mode trainable Gaussian state
settings.SEED = 7
gaussian = Gaussian(num_modes=2,
                    symplectic_trainable=True,
                    cutoffs=[cutoff, 4])

def output():
    return gaussian << Fock(3, modes=[1])

def cost_fn():
    fid = fidelity(normalize(output()), cat_target)
    if fid > 0.99:
        prob = output().probability
        return 1 - fid - prob
    else:
        return 1 - fid

opt = Optimizer(symplectic_lr = 0.2)
opt.minimize(cost_fn, by_optimizing=[gaussian],
             max_steps=150)
```

This cost function includes the probability of the state when the fidelity is above 99%.

It should be observed that the Fock space cutoff selected for this optimization (100) was quite large. However, the choice was deliberately made to demonstrate the speed of our method: the cat state optimization took approximately three seconds to complete on an M1 MacBook Air using MrMustard version 0.5.0.

### 6.2.2 Cubic phase state

The ideal cubic phase state is given by the cubic phase gate applied to a momentum squeezed vacuum state $V(\gamma) = e^{i\frac{\gamma}{3\hbar}x^3}|0\rangle_p$, which has infinite energy. We target the finite-energy version $e^{i\frac{\gamma}{3\hbar}x^3}S(r)|0\rangle$ for $\gamma = 0.3\hbar$ and $r = -1$:

$$|\text{cubic}\rangle = e^{\frac{1}{10}ix^3}e^{\frac{1}{2}(a^{\dagger 2}-a^2)}|0\rangle. \tag{145}$$

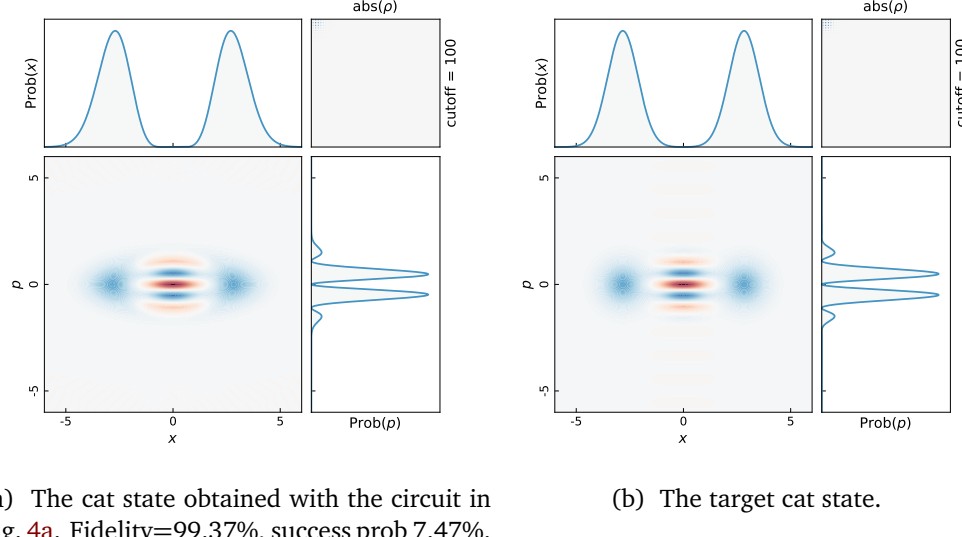

(a) The cat state obtained with the circuit in Fig. 4a. Fidelity=99.37%, success prob 7.47%.

(b) The target cat state.

Figure 5: Optimized cat state and the target state.

This target state is shown in Fig. 6b. We follow a different optimization strategy than the one we followed for the cat state. Specifically, even though we target the measurement of 16,16 photons in the last two modes, we find it is beneficial to optimize lower photon number measurements first and work our way up to the target measurement (16,16 in this case), re-optimizing the 3-mode Gaussian state for each step. We find a solution with 99.00% fidelity and probability = 0.06%, which we report in the Appendix.

The symplectic matrix of the 3-mode Gaussian transformation and the displacements are reported here below. Note that at this stage, as we didn't commit to a specific circuit design, but rather we optimized the Gaussian symplectic matrix, we still have a relative amount of flexibility in realizing this Gaussian transformation in the way that is most convenient given some constraints (e.g. the order of the gates that the hardware allows for).

## 7 Extensions to linear combinations of Gaussians

While the set of Gaussian states is rather restrictive, many non-Gaussian states of interest, such as cat states, Gottesman-Kitaev-Preskill (GKP) states [65], or Fock states, can be exactly or approximately expanded as linear combinations of Gaussians in phase space [36, 66]. This representation has the nice property that any Gaussian channel can act on these states directly in phase space, i.e., without requiring to write their Fock representation explicitly. Because of linearity, we can simply obtain the Fock representation of any states expressible as a linear combination of Gaussians by obtaining the Fock representation of each Gaussian component. This argument is equally valid for pure and mixed states. For the case of pure states, it is important to correctly account for the global phase as described in the previous sections. This phase will be important for states for which the coefficients $c_\psi$ have non-trivial dependence on the displacement and squeezing that describes each individual component, as it is apparent in squeezed-comb states defined as [67]

$$|0_{\text{Comb}}\rangle = \frac{1}{\mathcal{N}_{\text{Comb}}} \sum_{n=1}^{N} |\psi_n\rangle, \quad |\psi_n\rangle = \mathcal{D}(\bar{q}_n)\mathcal{S}(r)|0\rangle, \tag{146}$$

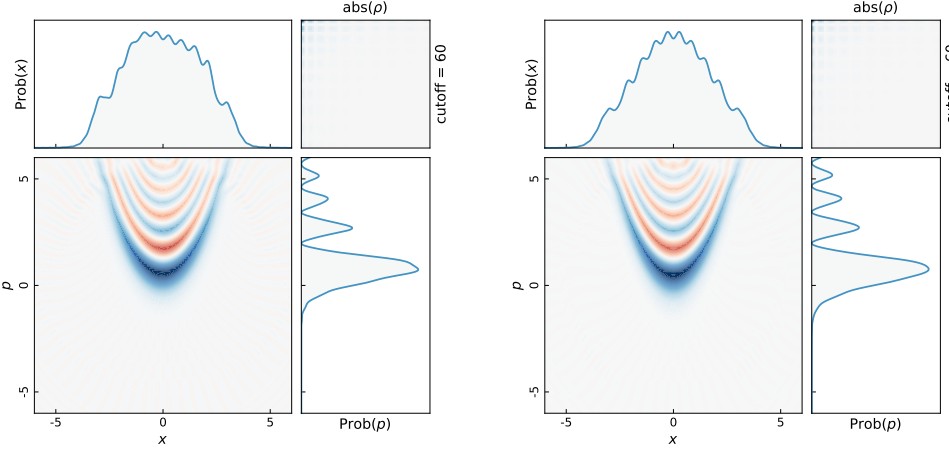

(a) The cubic phase state obtained with the circuit in Fig. 4b. Fidelity=99%, success prob 0.06%.

(b) The target cubic phase state.

Figure 6: Optimized cubic phase state and the target state.

where recall $\mathcal{D}(\cdot)$ and $\mathcal{S}(\cdot)$ are the single-mode displacement and squeezing operator defined in Sec. 2.3, $q_n = -(N+1)(d/2) + nd$. Note that squeezed-comb states have as limit both cat states (when the squeezing parameters are zero and $N = 2$) and GKP states (when $r > 0$ and $N$ is large). Note that each element in the linear combination will have a non-trivial phase that appears in a linear superposition and thus cannot be factored out as a global phase, making clear the relevance of the results in Sec. 4.

Consider now the density matrix associated with the state above

$$\rho_{0_{\text{Comb}}} = |0_{\text{Comb}}\rangle\langle 0_{\text{Comb}}| \tag{147}$$

$$= \sum_{n=1}^{N} |\psi_n\rangle\langle\psi_n| + \sum_{n=1}^{N} \sum_{m=1,m\neq n}^{N} |\psi_n\rangle\langle\psi_m|. \tag{148}$$

On the one hand, the "diagonal" terms $|\psi_n\rangle\langle\psi_n|$ correspond to positive semi-definite operators with Gaussian characteristic functions. On the other hand, the "off-diagonal" terms $|\psi_n\rangle\langle\psi_m|$ do not represent positive semi-definite operators but they still have complex-Gaussian characteristic functions as shown in Appendix A of Ref. [36]. This implies that the recursion relations derived in this manuscript still hold for each term in the equation above. Finally, note that certain non-Gaussian operations can also be described in terms of linear combinations of Gaussian. The Kerr gate

$$K(\kappa) = \exp\left[i\kappa a^\dagger a^\dagger a a\right], \tag{149}$$

with parameter $\kappa = \pi/m$ can be expanded as a linear combination of rotation gates [68]. Thus the methods we developed, including the global phase will be important when composing this gate with other Gaussian operators with non-trivial phase terms $c$ so as to achieve universality.

## 8 Conclusion

In this work we have presented a linear recurrence relation that connects the phase space and the Fock space representations of Gaussian pure and mixed states, as well as Gaussian

unitary and non-unitary transformations. While working with Gaussian gates within the phase space representation is easily achieved using symplectic algebra, it is valuable to implement fast numerical simulations in Fock representation, in order to include non-Gaussian effects. Moreover, the recurrence relation is exact and differentiable, which enables accurate gradient computations and gradient-based optimization.

Since the covariance matrix of Gaussian objects is parametrized by symplectic matrices that live in a Riemannian manifold, a geodesic-based optimization method is proposed in this paper. We show some optimization examples using the open-source library MrMustard, where we implemented our methods. In particular, we optimized the adjacency matrix of a high-dimensional Gaussian Boson Sampling instance with 216 modes directly in phase space to highlight the Euclidean optimization functionality of our library.

We then obtained new circuits to generate mesoscopic cat states with unprecedented success probability. On the theory side, we also showed how to keep track of the global phase induced by Gaussian unitary transformations. This paves the way to simulate and optimize non-Gaussian objects by writing them as linear combinations of Gaussians [36]. Dealing with non-Gaussian simulation and optimization is a significant challenge in the optical information processing community [12, 69]. Our methods offer a promising avenue to address this challenge.

## Acknowledgments

N.Q. thanks R. Chadwick and T. Kalajdzievski for valuable discussions.

**Funding information** N.Q. thanks the Ministère de l'Économie et de l'Innovation du Québec and the Natural Sciences and Engineering Research Council of Canada for financial support.

## A  Review of the symplectic formalism

The real symplectic group is defined as

$$\mathrm{Sp}(2n, \mathbb{R}) = \{ S \in \mathbb{R}^{2n \times 2n} | S \Omega S^T = \Omega \}, \tag{A.1}$$

where $\Omega$ is defined in Eq. (8).

Some properties of this group:

$$\Omega \in \mathrm{Sp}(2n, \mathbb{R}), \tag{A.2}$$

$$\Omega^{-1} = \Omega^T = -\Omega \in \mathrm{Sp}(2n, \mathbb{R}), \tag{A.3}$$

$$S^{-1} = -\Omega S^T \Omega \in \mathrm{Sp}(2n, \mathbb{R}). \tag{A.4}$$

A real symplectic matrix $S$ can be decomposed as

$$S = O_1 \Lambda O_2, \tag{A.5}$$

with $O_1, O_2 \in C(n)$ and

$$\lambda = \Lambda \oplus \Lambda^{-1}, \tag{A.6}$$

with $\Lambda = \mathrm{diag}(\lambda_1, \ldots, \lambda_n)$ and $\lambda_j \geq 1, \forall j \in [1, \ldots, n]$. $C(n)$ denotes the compact subgroup and $C(n) = \mathrm{Sp}_{2n, \mathbb{R}} \cap O(2n)$. It means that any symplectic matrix can be decomposed into a diagonal and positive semi-definite matrix $\Lambda$ with two orthogonal groups $O_1$ and $O_2$, which stands for the passive transformation (interferometer).

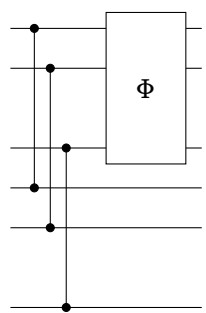

Figure 7: $2M$-mode circuit for implementing the Choi-Jamiołkowski duality. $\Phi$ is the channel that is applied on the first half $M$ modes and the two dots represent a two-mode squeezing operator connecting two modes: one comes from the first $M$ modes and the other one comes from the second $M$ modes.

# B  Choi-Jamiołkowski duality

In this section, we employ the Choi-Jamiołkowski duality [38,41,42] to reduce the calculation of the matrix elements of an arbitrary Gaussian channel in $M$ to the calculation of the matrix element of a Gaussian state with $2M$. We first consider a collection of systems with arbitrary, but identical, dimensionality $N$.

We write the state right before the channel $\Phi$ is applied to the first half of the modes in Fig. 7 as

$$|\Psi\rangle = \sqrt{\mathcal{N}} \sum_{n=0}^{N-1} \tau^n |n\rangle \otimes |n\rangle, \tag{B.1}$$

where $\sum_{n=0}^{N-1} \equiv \sum_{n_1=0}^{N-1} \cdots \sum_{n_M=0}^{N-1}$, $\mathcal{N}$ is a normalization constant to be determined in a moment and $\tau$ is the squeezing parameter of the two-mode squeezing operator connecting the first $M$ modes and the second $M$ modes. The density matrix of the state $|\Psi\rangle$ is simply

$$|\Psi\rangle\langle\Psi| = \mathcal{N} \sum_{m=0}^{N-1} \sum_{n=0}^{N-1} \tau^{n+m} |n\rangle\langle m| \otimes |n\rangle\langle m|. \tag{B.2}$$

We can now write the output of the circuit after the application of the channel $\Phi$ as

$$\varrho = (\Phi \otimes \mathbb{I})[|\Psi\rangle\langle\Psi|] = \mathcal{N} \sum_{m=0}^{N-1} \sum_{n=0}^{N-1} \tau^{n+m} \Phi[|n\rangle\langle m|] \otimes |n\rangle\langle m|. \tag{B.3}$$

We can premultiply the equation above by $\langle i| \otimes \langle j|$ and postmultiply by $|k\rangle \otimes |l\rangle$ to obtain

$$(\langle i| \otimes \langle j|)\varrho(|k\rangle \otimes |l\rangle) = \mathcal{N}\, \tau^{j+l}\langle i|(\Phi[|j\rangle\langle l|])|k\rangle. \tag{B.4}$$

In finite-dimensional systems it is convenient to pick $\tau = (1,\ldots,1)$ and the normalization $\mathcal{N}$ is simply given by the dimensionality of the system $N^M$. For infinite dimensional systems, if one were to try to pick the same normalization as for a finite-dimensional, one would obtain a non-normalizable state $|\Psi\rangle$. Thus it is convenient to pick $\tau = (\tau,\ldots,\tau)$ with $\tau = \tanh t < 1$ and then

$$\mathcal{N} = (1-\tau^2)^M = (1-\tanh^2 t)^M, \tag{B.5}$$

$$\tau^{l+j} = (\tanh t)^{\sum_{i=1}^M l_i + j_i}. \tag{B.6}$$

For a rigorous justification of this derivation see sec 5.5 of Serafini [38]. Now consider the case where the channel $\Phi$ is Gaussian parametrized by

$$X = \begin{bmatrix} X_{qq} & X_{qp} \\ X_{pq} & X_{pp} \end{bmatrix}, \quad Y = \begin{bmatrix} Y_{qq} & Y_{qp} \\ Y_{pq} & Y_{pp} \end{bmatrix}, \quad d = \begin{bmatrix} d_q \\ d_p \end{bmatrix}. \tag{B.7}$$

Then the output state is also Gaussian since the input state to the channel is nothing but one-half of a two-mode squeezed state. In this case, we can write the quadrature covariance matrix and vector of means of the output state as

$$V = \tilde{X}\mathcal{T}(t)\left(\frac{\hbar}{2}\mathbb{1}_{4M}\right)\mathcal{T}(t)^T\tilde{X}^T + \tilde{Y} = \frac{\hbar}{2}\tilde{X}\mathcal{T}(2t)\tilde{X}^T + \tilde{Y}, \qquad \bar{r} = \begin{bmatrix} d_q \\ 0 \\ d_p \\ 0 \end{bmatrix}, \tag{B.8}$$

where

$$\tilde{X} = \begin{bmatrix} X_{qq} & 0_M & X_{qp} & 0_M \\ 0_M & \mathbb{1}_M & 0_M & 0_M \\ X_{pq} & 0_M & X_{pp} & 0_M \\ 0_M & 0_M & 0_M & \mathbb{1}_M \end{bmatrix}, \quad \tilde{Y} = \begin{bmatrix} Y_{qq} & 0_M & Y_{qp} & 0_M \\ 0_M & 0_M & 0_M & 0_M \\ Y_{pq} & 0_M & Y_{pp} & 0_M \\ 0_M & 0_M & 0_M & 0_M \end{bmatrix}, \tag{B.9}$$

$$\mathcal{T}(t) = \begin{bmatrix} \cosh t\, \mathbb{1}_M & \sinh t\, \mathbb{1}_M & 0_M & 0_M \\ \sinh t\, \mathbb{1}_M & \cosh t\, \mathbb{1}_M & 0_M & 0_M \\ 0_M & 0_M & \cosh t\, \mathbb{1}_M & -\sinh t\, \mathbb{1}_M \\ 0_M & 0_M & -\sinh t\, \mathbb{1}_M & \cosh t\, \mathbb{1}_M \end{bmatrix}, \tag{B.10}$$

and we used the fact that $\mathcal{T}(t)\mathcal{T}(t)^T = \mathcal{T}(t)\mathcal{T}(t) = \mathcal{T}(2t)$.

In the next appendix, we show that we can associate with the $2M$-Gaussian Choi-Jamiołkowski state the following quantities

$$A_\varrho = E(t)A_\Phi E(t), \tag{B.11}$$

$$A_\Phi = P_{2M}R\begin{bmatrix} \mathbb{1}_{2M} - \xi^{-1} & \xi^{-1}X \\ X^T\xi^{-1} & \mathbb{1}_{2M} - X^T\xi^{-1}X \end{bmatrix}R^\dagger \tag{B.12}$$

$$= P_{2M}R\left(\mathbb{1}_{4M} - \begin{bmatrix} \xi^{-1} & -\xi^{-1}X \\ -X^T\xi^{-1} & X^T\xi^{-1}X \end{bmatrix}\right)R^\dagger, \tag{B.13}$$

$$b_\varrho = E(t)b_\Phi, \tag{B.14}$$

$$b_\Phi = \frac{1}{\sqrt{\hbar}}R^*\begin{bmatrix} \xi^{-1}d \\ -X^T\xi^{-1}d \end{bmatrix}, \tag{B.15}$$

$$c_\varrho = (1 - \tanh^2 t)^M c_\Phi, \tag{B.16}$$

$$c_\Phi = \frac{\exp\left[-\frac{1}{2\hbar}d^T\xi^{-1}d\right]}{\sqrt{\det(\xi)}}, \tag{B.17}$$

where $E(t) = \mathbb{1}_M \oplus (\tanh t \mathbb{1}_M) \oplus \mathbb{1}_M \oplus (\tanh t \mathbb{1}_M)$, $P_M = \begin{bmatrix} 0_M & \mathbb{1}_M \\ \mathbb{1}_M & 0_M \end{bmatrix}$ and

$$R = \frac{1}{\sqrt{2}} \begin{bmatrix} \mathbb{1}_M & i\mathbb{1}_M & 0_M & 0_M \\ 0_M & 0 & \mathbb{1}_M & -i\mathbb{1}_M \\ \mathbb{1}_M & -i\mathbb{1}_M & 0_M & 0_M \\ 0_M & 0_M & \mathbb{1}_M & i\mathbb{1}_M \end{bmatrix}, \quad \xi = \frac{1}{2}\left(\mathbb{1}_{2M} + XX^T + \frac{2Y}{\hbar}\right). \tag{B.18}$$

Note that $\xi$ is nothing but the $qp$-Husimi covariance matrix (in units where $\hbar = 1$) of the state obtained by sending the $M$ mode vacuum state in the process specified by $X$ and $Y$. Note that in general, given a covariance matrix $V$ one can always construct (a non-unique) channel that when applied to the vacuum produces the state with covariance matrix $V$. To this end recall that the Williamson decompositions states that any valid quantum covariance matrix can be written as $V = S(V_{\text{vac}} + V_{\text{noise}})S^T$ with $S$ symplectic, $V_{\text{vac}}$ is the covariance matrix of the vacuum and $V_{\text{noise}}$ positive semidefinite. The channel with $X = SO$ and $Y = SV_{\text{noise}}S^T$ with $O$ symplectic and orthogonal but otherwise arbitrary prepares the sought after state when applied on vacuum.

With these results we can write

$$(\langle i| \otimes \langle j|)\varrho(|k\rangle \otimes |l\rangle) = c_\varrho \times \frac{G^{A_\rho}_{k \oplus l \oplus i \oplus j}(b_\varrho)}{\sqrt{i!j!k!l!}}. \tag{B.19}$$

Now we recall a fundamental property that multidimensional Hermite polynomials inherit from loop-hafnians [44], namely that if $E = \oplus_{i=1}^{\ell} E_i$ is a diagonal matrix then

$$G^{EAE}_n(Eb) = \left(\prod_{i=1}^{\ell} E_i^{n_i}\right) G^A_n(b). \tag{B.20}$$

We can use the definitions from Eq. (B.11) to Eq. (B.17) together with the Eq. (B.4) and the relation Eq. (B.19) to find

$$\langle i|(\Phi[|j\rangle\langle l|])|k\rangle = \frac{(\langle i| \otimes \langle j|)\varrho(|k\rangle \otimes |l\rangle)}{\mathcal{N} \, \tau^{j+l}} = \frac{c_\varrho}{\mathcal{N} \, \tau^{j+l}} \times \frac{G^{A_\rho}_{k \oplus l \oplus i \oplus j}(b_\varrho)}{\sqrt{i!j!k!l!}} = c_\Phi \times \frac{G^{A_\Phi}_{k \oplus l \oplus i \oplus j}(b_\Phi)}{\sqrt{i!j!k!l!}}, \tag{B.21}$$

which allows us to find the matrix elements of the channel without any reference to the specific amount of squeezing used to create the two-mode squeezed vacuum.

## C Description of the Choi-Jamiołkowski duality in phase-space

The (complex) covariance matrix $\sigma$ of the Gaussian state obtained by sending $M$ halves of $M$ two-mode squeezed vacuum states through the channel $\Phi$ is given by

$$\sigma = W\left(\frac{1}{2}\tilde{X}\mathcal{T}(2t)\tilde{X}^T + \frac{\tilde{Y}}{\hbar}\right)W^\dagger. \tag{C.1}$$

Note that $(\mathcal{T}(t))^T = \mathcal{T}(t)$ is symmetric, $\tilde{X}$ is symplectic if $X = \begin{bmatrix} X_{qq} & X_{qp} \\ X_{pq} & X_{pp} \end{bmatrix}$ is symplectic and $W$ is unitary. Let

$$Q' = \left(\frac{\mathbb{1}_{4M}}{2} + \frac{1}{2}\tilde{X}\mathcal{T}(2t)\tilde{X}^T + \frac{\tilde{Y}}{\hbar}\right), \tag{C.2}$$

then $(\sigma + \frac{\mathbb{1}_{4M}}{2})^{-1} = W(Q')^{-1}W^{\dagger}$. Now we define

$$Q = LQ'L^{T}, \tag{C.3}$$

with

$$L = \begin{bmatrix} \mathbb{1}_M & 0_M & 0_M & 0_M \\ 0_M & 0_M & \mathbb{1}_M & 0_M \\ 0_M & \mathbb{1}_M & 0_M & 0_M \\ 0_M & 0_M & 0_M & \mathbb{1}_M \end{bmatrix}. \tag{C.4}$$

Then we have that $Q^{-1} = L(Q')^{-1}L^{T}$, which implies that $L^{T}Q^{-1}L = (Q')^{-1}$. So calculating $Q^{-1}$ gives $(Q')^{-1}$ and therefore $(\sigma + \frac{\mathbb{1}_{4M}}{2})^{-1}$.

Expressing $Q$ as a block matrix $Q = \begin{bmatrix} A & B \\ C & D \end{bmatrix}$, we can write $Q^{-1}$ using Schur complements as [38]

$$Q^{-1} = \begin{bmatrix} \xi^{-1} & -\xi^{-1}BD^{-1} \\ -D^{-1}C\xi^{-1} & D^{-1} + D^{-1}C\xi^{-1}BD^{-1} \end{bmatrix}, \tag{C.5}$$

where $\xi = A - BD^{-1}C$. The blocks $A$, $B$, $C$, and $D$, are given by

$$A = \frac{Y}{\hbar} + \frac{\mathbb{1}_{2M}}{2} + \frac{1}{2}\cosh(2t)XX^{T}, \tag{C.6}$$

$$B = \frac{1}{2}\sinh 2t \begin{bmatrix} X_{qq} & -X_{qp} \\ X_{pq} & -X_{pp} \end{bmatrix} = \frac{1}{2}\sinh 2t XZ, \tag{C.7}$$

$$C = \frac{1}{2}\sinh 2t \begin{bmatrix} X_{qq}^{T} & X_{pq}^{T} \\ -X_{qp}^{T} & -X_{pp}^{T} \end{bmatrix} = B^{T} = \frac{1}{2}\sinh 2t ZX^{T}, \tag{C.8}$$

$$D = \cosh^{2}(t) \begin{bmatrix} \mathbb{1}_M & 0_M \\ 0_M & \mathbb{1}_M \end{bmatrix}, \tag{C.9}$$

where $Z = \begin{bmatrix} \mathbb{1}_M & 0_M \\ 0_M & -\mathbb{1}_M \end{bmatrix}$. We now use these to calculate the blocks of $Q^{-1}$ starting with $\xi$,

$$\xi = A - BD^{-1}C = \frac{1}{2}\left(\mathbb{1}_{2M} + XX^{T} + \frac{2Y}{\hbar}\right) = \xi^{T}, \tag{C.10}$$

which turns out to be *independent* of $t$. Next, we find

$$-\xi^{-1}BD^{-1} = -\tanh(t)\xi^{-1}XZ, \tag{C.11}$$

$$-D^{-1}C\xi^{-1} = -\tanh(t)ZX^{T}\xi^{-1}. \tag{C.12}$$

Finally, the bottom right block, which can be simplified by substituting the other three blocks, is given by

$$D^{-1} + D^{-1}C\xi^{-1}BD^{-1} = \left(1 - \tanh^{2}(t)\right)\mathbb{1}_{2M} + \tanh^{2}(t)ZX^{T}\xi^{-1}XZ \tag{C.13}$$

$$= \mathbb{1}_{2M} + \tanh^{2}(t)Z\left(X^{T}\xi^{-1}X - \mathbb{1}_{2M}\right)Z. \tag{C.14}$$

Putting these blocks together, we get the expanded form of $Q^{-1}$

$$Q^{-1} = \begin{bmatrix} \xi^{-1} & -\tanh(t)\xi^{-1}XZ \\ -\tanh(t)ZX^T\xi^{-1} & \mathbb{1}_{2M} + \tanh^2(t)Z\left(X^T\xi^{-1}X - \mathbb{1}_{2M}\right)Z \end{bmatrix}. \tag{C.15}$$

Now with the form of the inverse known, we can multiply by the remaining matrices to get the final form of $\sigma_+^{-1} = (\sigma + \frac{\mathbb{1}_{4M}}{2})^{-1} = WL^TQ^{-1}LW^\dagger$, with $L$ as in Eq. (C.4)

We can now go back and write the quantity of interest

$$\mathbb{1}_{4M} - \left(\sigma + \frac{\mathbb{1}_{4M}}{2}\right)^{-1} = WL^T\left(\mathbb{1}_{4M} - \begin{bmatrix} \xi^{-1} & -\tanh(t)\xi^{-1}XZ \\ -\tanh(t)ZX^T\xi^{-1} & \mathbb{1}_{2M} + \tanh^2(t)Z\left(X^T\xi^{-1}X - \mathbb{1}_{2M}\right)Z \end{bmatrix}\right)LW^\dagger \tag{C.16}$$

$$= WL^T\begin{bmatrix} \mathbb{1}_{2M} - \xi^{-1} & \tanh(t)\xi^{-1}XZ \\ \tanh(t)ZX^T\xi^{-1} & \tanh^2(t)Z\left[\mathbb{1}_{2M} - X^T\xi^{-1}X\right]Z \end{bmatrix}LW^\dagger. \tag{C.17}$$

Defining the matrix $F = \begin{bmatrix} \mathbb{1}_{2M} & 0_{2M} \\ 0_{2M} & Z\tanh(t) \end{bmatrix}$, we can rewrite the last equation as

$$\mathbb{1}_{4M} - \left(\sigma + \frac{\mathbb{1}_{4M}}{2}\right)^{-1} = WL^TF\begin{bmatrix} \mathbb{1}_{2M} - \xi^{-1} & \xi^{-1}X \\ X^T\xi^{-1} & \mathbb{1}_{2M} - X^T\xi^{-1}X \end{bmatrix}F^TLW^\dagger \tag{C.18}$$

$$= E(t)R\begin{bmatrix} \mathbb{1}_{2M} - \xi^{-1} & \xi^{-1}X \\ X^T\xi^{-1} & \mathbb{1}_{2M} - X^T\xi^{-1}X \end{bmatrix}R^\dagger E(t), \tag{C.19}$$

where we noted that $WL^TF = WLF = E(t)R$ (cf. Eq. (B.18)). To arrive at the expression for $A_\rho$ we simply note $[E(t), P_{2M}] = 0$.

We would also like to find

$$b_\varrho = \left(\sigma_+^{-1}\bar{\mu}\right)^* = \left(WLQ^{-1}LW^\dagger\left[\frac{1}{\sqrt{\hbar}}W\bar{r}\right]\right)^* = \frac{1}{\sqrt{\hbar}}\left(WLQ^{-1}L\bar{r}\right)^* = \frac{1}{\sqrt{\hbar}}(WL)^*Q^{-1}\begin{bmatrix} d \\ 0 \end{bmatrix} \tag{C.20}$$

$$= \frac{1}{\sqrt{\hbar}}(WL)^*\begin{bmatrix} \xi^{-1}d \\ -\tanh t ZX^T\xi^{-1}d \end{bmatrix} = \frac{1}{\sqrt{\hbar}}(WLF)^*\begin{bmatrix} \xi^{-1}d \\ -X^T\xi^{-1}d \end{bmatrix} = \frac{1}{\sqrt{\hbar}}E(t)R^*\begin{bmatrix} \xi^{-1}d \\ -X^T\xi^{-1}d \end{bmatrix}. \tag{C.21}$$

Finally, we can obtain

$$c_\varrho = (\langle\mathbf{0}| \otimes \langle\mathbf{0}|)\varrho(|\mathbf{0}\rangle \otimes |\mathbf{0}\rangle) = \mathcal{N}\langle\mathbf{0}|(\Phi[|\mathbf{0}\rangle\langle\mathbf{0}|])|\mathbf{0}\rangle = \mathcal{N}c_\Phi. \tag{C.22}$$

The Husimi covariance matrix of the state $\Phi[|\mathbf{0}\rangle\langle\mathbf{0}|]$ is simply $\hbar\xi$ and its vector of means is $d$ and thus we can write

$$\langle\mathbf{0}|(\Phi[|\mathbf{0}\rangle\langle\mathbf{0}|])|\mathbf{0}\rangle = \frac{\exp\left[-\frac{1}{2}d^T(\hbar\xi)^{-1}d\right]}{\sqrt{\det(\xi)}}. \tag{C.23}$$

## D Unitary processes

Now consider a unitary process. In this case we know that $Y = 0$ and that $X = S \in Sp_{2M}$ where $Sp_{2M}$ is the Symplectic group. Since $S$ is symplectic, then we can write a symplectic

singular-value decomposition (also known as a Bloch-Messiah or Euler decomposition [70])

$$
S = \begin{bmatrix} \Re(U_1) & -\Im(U_1) \\ \Im(U_1) & \Re(U_1) \end{bmatrix} \underbrace{\begin{bmatrix} e^{-r} & 0_M \\ 0_M & e^{r} \end{bmatrix}}_{\equiv \lambda} \begin{bmatrix} \Re(U_2) & -\Im(U_2) \\ \Im(U_2) & \Re(U_2) \end{bmatrix} = O_1 \lambda O_2 , \tag{D.1}
$$

where $U_1, U_2$ are $M \times M$ unitaries and $r = \oplus_{i=1}^M r_i$ represents squeezing.

We can now calculate the Schur complement to find

$$
\xi = \frac{1}{2} \left( \mathbb{1}_{2M} + S S^T \right) , \tag{D.2}
$$

$$
\xi^{-1} = 2 O_1 \frac{\mathbb{1}_{2M}}{\mathbb{1}_{2M} + \lambda^2} O_1^T , \tag{D.3}
$$

and then we find

$$
\begin{bmatrix} \mathbb{1}_{2M} - \xi^{-1} & \xi^{-1} X \\ X^T \xi^{-1} & \mathbb{1}_{2M} - X^T \xi^{-1} X \end{bmatrix} = \begin{bmatrix} O_1 \frac{\lambda^2 - \mathbb{1}_{2M}}{\lambda^2 + \mathbb{1}_{2M}} O_1^T & O_1 \frac{2\lambda}{\lambda^2 + \mathbb{1}_{2M}} O_2 \\ O_2^T \frac{2\lambda}{\lambda^2 + \mathbb{1}_{2M}} O_1^T & -O_2^T \frac{\lambda^2 - \mathbb{1}_{2M}}{\lambda^2 + \mathbb{1}_{2M}} O_2 \end{bmatrix} \tag{D.4}
$$

$$
= \begin{bmatrix} O_1 & 0_{2M} \\ 0_{2M} & O_2^T \end{bmatrix} \begin{bmatrix} \frac{\lambda^2 - \mathbb{1}_{2M}}{\lambda^2 + \mathbb{1}_{2M}} & \frac{2\lambda}{\lambda^2 + \mathbb{1}_{2M}} \\ \frac{2\lambda}{\lambda^2 + \mathbb{1}_{2M}} & -\frac{\lambda^2 - \mathbb{1}_{2M}}{\lambda^2 + \mathbb{1}_{2M}} \end{bmatrix} \begin{bmatrix} O_1^T & 0_{2M} \\ 0_{2M} & O_2 \end{bmatrix} . \tag{D.5}
$$

Note that

$$
\frac{\lambda^2 - \mathbb{1}_{2M}}{\lambda^2 + \mathbb{1}_{2M}} = \begin{bmatrix} -\tanh r & 0_M \\ 0_M & \tanh r \end{bmatrix} , \qquad \frac{2\lambda}{\lambda^2 + \mathbb{1}_{2M}} = \begin{bmatrix} \operatorname{sech} r & 0_M \\ 0_M & \operatorname{sech} r \end{bmatrix} . \tag{D.6}
$$

We can now calculate

$$
R \begin{bmatrix} \mathbb{1}_{2M} - \xi^{-1} & \xi^{-1} X \\ X^T \xi^{-1} & \mathbb{1}_{2M} - X^T \xi^{-1} X \end{bmatrix} R^\dagger
$$

$$
= R \begin{bmatrix} O_1 & 0_M \\ 0_M & O_2^T \end{bmatrix} R^\dagger R \begin{bmatrix} \frac{\lambda^2 - \mathbb{1}_{2M}}{\lambda^2 + \mathbb{1}_{2M}} & \frac{2\lambda}{\lambda^2 + \mathbb{1}_{2M}} \\ \frac{2\lambda}{\lambda^2 + \mathbb{1}_{2M}} & -\frac{\lambda^2 - \mathbb{1}_{2M}}{\lambda^2 + \mathbb{1}_{2M}} \end{bmatrix} R^\dagger R \begin{bmatrix} O_1^T & 0_M \\ 0_M & O_2 \end{bmatrix} R^\dagger \tag{D.7}
$$

$$
= \begin{bmatrix} U_1 & 0_M & 0_M & 0_M \\ 0_M & U_2^T & 0_M & 0_M \\ 0_M & 0_M & U_1^* & 0_M \\ 0_M & 0_M & 0_M & U_2^\dagger \end{bmatrix} \begin{bmatrix} 0_M & 0_M & -\tanh r & \operatorname{sech} r \\ 0_M & 0_M & \operatorname{sech} r & \tanh r \\ -\tanh r & \operatorname{sech} r & 0_M & 0_M \\ \operatorname{sech} r & \tanh r & 0_M & 0_M \end{bmatrix} \begin{bmatrix} U_1 & 0_M & 0_M & 0_M \\ 0_M & U_2^T & 0_M & 0_M \\ 0_M & 0_M & U_1^* & 0_M \\ 0_M & 0_M & 0_M & U_2^\dagger \end{bmatrix}^\dagger \tag{D.8}
$$

$$
= - \begin{bmatrix} 0_M & A_U \\ A_U^* & 0_M \end{bmatrix} , \tag{D.9}
$$

where

$$
A_U = \begin{bmatrix} U_1 & 0_M \\ 0_M & U_2^T \end{bmatrix} \begin{bmatrix} \tanh r & -\operatorname{sech} r \\ -\operatorname{sech} r & -\tanh r \end{bmatrix} \begin{bmatrix} U_1 & 0_M \\ 0_M & U_2^T \end{bmatrix}^T = A_U^T . \tag{D.10}
$$

We can also explicitly calculate $\boldsymbol{b}_\Phi$ to find

$$
\boldsymbol{b}_\Phi = \frac{1}{\sqrt{2\hbar}}
\begin{bmatrix}
\boldsymbol{d}_q - i\boldsymbol{d}_p + \boldsymbol{U}_1^* \tanh r \boldsymbol{U}_1^\dagger (\boldsymbol{d}_q + i\boldsymbol{d}_p) \\
-\boldsymbol{U}_2^\dagger \operatorname{sech} r \boldsymbol{U}_1^T (\boldsymbol{d}_q - i\boldsymbol{d}_p) \\
\boldsymbol{d}_q + i\boldsymbol{d}_p + \boldsymbol{U}_1 \tanh r \boldsymbol{U}_1^T (\boldsymbol{d}_q - i\boldsymbol{d}_p) \\
-\boldsymbol{U}_2^T \operatorname{sech} r \boldsymbol{U}_1^\dagger (\boldsymbol{d}_q + i\boldsymbol{d}_p)
\end{bmatrix}
=
\begin{bmatrix}
\boldsymbol{\alpha}^* + \boldsymbol{U}_1^* \tanh r \boldsymbol{U}_1^\dagger \boldsymbol{\alpha} \\
-\boldsymbol{U}_2^\dagger \operatorname{sech} r \boldsymbol{U}_1^T \boldsymbol{\alpha}^* \\
\boldsymbol{\alpha} + \boldsymbol{U}_1 \tanh r \boldsymbol{U}_1^T \boldsymbol{\alpha}^* \\
-\boldsymbol{U}_2^T \operatorname{sech} r \boldsymbol{U}_1^\dagger \boldsymbol{\alpha}
\end{bmatrix}
=
\begin{bmatrix}
\boldsymbol{b}_U^* \\
\boldsymbol{b}_U
\end{bmatrix}
, \quad \text{(D.11)}
$$

where we wrote $\boldsymbol{d} = \begin{bmatrix} \boldsymbol{d}_q \\ \boldsymbol{d}_p \end{bmatrix}$ and introduced $\boldsymbol{\alpha} = \frac{1}{\sqrt{2\hbar}} (\boldsymbol{d}_q + i\boldsymbol{d}_p)$. Finally, we find for the scalar $c$

$$
\begin{aligned}
c_\Phi &= \frac{\exp\left(-\frac{1}{4\hbar}\left[||\boldsymbol{d}||^2 + (\boldsymbol{d}_q^T - i\boldsymbol{d}_p^T)\boldsymbol{U}_1 \tanh r \boldsymbol{U}_1^T (\boldsymbol{d}_q - i\boldsymbol{d}_p) + \text{c.c.}\right]\right)}{\prod_{i=1}^M \cosh r_i} \\
&= \frac{\exp\left(-||\boldsymbol{\alpha}||^2 - \Re\left[\boldsymbol{\alpha}^\dagger \boldsymbol{U}_1 \tanh r \boldsymbol{U}_1^T \boldsymbol{\alpha}^*\right]\right)}{\prod_{i=1}^M \cosh r_i} = |c_U|^2 .
\end{aligned}
\quad \text{(D.12)}
$$

## E   Passive processes

Now consider the case of non-unitary passive process specified by a transfer matrix $\boldsymbol{T}$, $\boldsymbol{T}^\dagger \boldsymbol{T} \le \mathbb{1}_M$. For this process $\boldsymbol{X} = \begin{bmatrix} \Re(\boldsymbol{T}) & -\Im(\boldsymbol{T}) \\ \Im(\boldsymbol{T}) & \Re(\boldsymbol{T}) \end{bmatrix}$ and $\boldsymbol{Y} = \frac{\hbar}{2}\left(\mathbb{1}_{2M} - \boldsymbol{X}\boldsymbol{X}^T\right)$. Since the process is passive we know that $\boldsymbol{\xi} = \mathbb{1}_{2M}$. We can simplify the expression to obtain

$$
\boldsymbol{P}_{2M}\left[\mathbb{1}_{4M} - \left(\boldsymbol{\sigma} + \frac{\mathbb{1}_{4M}}{2}\right)^{-1}\right] = \boldsymbol{E}(t)
\begin{bmatrix}
0_M & \boldsymbol{T}^* & 0_M & 0_M \\
\boldsymbol{T}^\dagger & 0_M & 0_M & \mathbb{1}_M - \boldsymbol{T}^\dagger \boldsymbol{T} \\
0_M & 0_M & 0_M & \boldsymbol{T} \\
0_M & \mathbb{1}_M - \boldsymbol{T}^T \boldsymbol{T}^* & \boldsymbol{T}^T & 0_M
\end{bmatrix}
\boldsymbol{E}(t). \quad \text{(E.1)}
$$

Following the Choi-Jamiołkowski relation we gave in Eq. (B.11), the $\boldsymbol{A}_\Phi$ for the lossy interferometer is

$$
\boldsymbol{A}_\Phi =
\begin{bmatrix}
0_M & \boldsymbol{T}^* & 0_M & 0_M \\
\boldsymbol{T}^\dagger & 0_M & 0_M & \mathbb{1}_M - \boldsymbol{T}^\dagger \boldsymbol{T} \\
0_M & 0_M & 0_M & \boldsymbol{T} \\
0_M & \mathbb{1}_M - \boldsymbol{T}^T \boldsymbol{T}^* & \boldsymbol{T}^T & 0_M
\end{bmatrix}.
\quad \text{(E.2)}
$$

If we sandwich $\boldsymbol{A}_\Phi$ with a permutation matrix $\boldsymbol{P}_{4123}$, we would have:

$$
\boldsymbol{P}_{4123} \boldsymbol{A}_\Phi \boldsymbol{P}_{4123}^T =
\begin{bmatrix}
0_M & 0_M & \mathbb{1}_M - \boldsymbol{T}^\dagger \boldsymbol{T} & \boldsymbol{T}^\dagger \\
0_M & 0_M & \boldsymbol{T} & 0_M \\
\mathbb{1}_M - \boldsymbol{T}^T \boldsymbol{T}^* & \boldsymbol{T}^T & 0_M & 0_M \\
\boldsymbol{T}^* & 0_M & 0_M & 0_M
\end{bmatrix}.
\quad \text{(E.3)}
$$

To get the probability with a measurement of photon number pattern $\boldsymbol{j} = (j_1, \ldots, j_M)$ given the input $\boldsymbol{i} = (i_1, \ldots, i_M)$, we let $\boldsymbol{k} = \boldsymbol{i}, \boldsymbol{l} = \boldsymbol{j}$ in (77):

$$G_{i \oplus j \oplus i \oplus j}^{A_\Phi}(0) = G_{P_{4123}(i \oplus j \oplus i \oplus j)}^{P_{4123} A_\Phi P_{4123}^T}(0) = \frac{1}{i! j!} \mathrm{haf}(P_{4123} A_\Phi P_{4123}^T)_{j \oplus i \oplus j \oplus i} = \frac{1}{i! j!} \mathrm{perm} \begin{bmatrix} \mathbb{1}_M - T^\dagger T & T^\dagger \\ T & 0_M \end{bmatrix}_{j \oplus i}. \quad \text{(E.4)}$$

## F  Global phase of the composition of two Gaussian transformations

In this section, we find an expression for the global phase of the transformation obtained by applying two consecutive Gaussian transformations. To this end we find the triplet $A_{U_f}, b_{U_f}, c_{U_f}$ associated with the $Q$ function of the net transformation $\mathcal{U}_f = \mathcal{U}_1 \mathcal{U}_2$ in terms of the triplets $A, b, c$ specifying the Husimi functions of the transformations $\mathcal{U}_1$ and $\mathcal{U}_2$.

Eq. (23) of Ref. [22] shows that the Husimi $Q$ function of an arbitrary Gaussian unitary can be characterized by three quantities $C, \boldsymbol{\mu}$, and $\Sigma$. As we already know the relation between $(C, \boldsymbol{\mu}, \Sigma) = (c_U, b_U, -A_U)$, now we will rewrite the Husimi $Q$-function for an arbitrary Gaussian unitary as:

$$\langle \boldsymbol{\alpha}^* | \mathcal{U} | \boldsymbol{\beta} \rangle = \exp\left(-\tfrac{1}{2}\left[||\boldsymbol{\alpha}||^2 + ||\boldsymbol{\beta}||^2\right]\right) c_U \exp\left(b_U^T \boldsymbol{\nu} + \frac{1}{2} \boldsymbol{\nu}^T A_U \boldsymbol{\nu}\right), \quad \text{where} \quad \boldsymbol{\nu} = \begin{bmatrix} \boldsymbol{\alpha} \\ \boldsymbol{\beta} \end{bmatrix}. \quad \text{(F.1)}$$

We first compose the two transformations and then insert the resolution of the identity $\frac{1}{\pi^M} \int_{\mathbb{C}^M} d^{2M} \boldsymbol{\alpha} |\boldsymbol{\alpha}\rangle\langle\boldsymbol{\alpha}| = I$ to find:

$$\langle \boldsymbol{\beta}^* | \mathcal{U}_1 \mathcal{U}_2 | \boldsymbol{\beta}' \rangle = \langle \boldsymbol{\beta}^* | \mathcal{U}_1 I \mathcal{U}_2 | \boldsymbol{\beta}' \rangle = \frac{1}{\pi^M} \int_{\mathbb{C}^M} d^{2M} \boldsymbol{\alpha} \langle \boldsymbol{\beta}^* | \mathcal{U}_1 | \boldsymbol{\alpha} \rangle \langle \boldsymbol{\alpha} | \mathcal{U}_2 | \boldsymbol{\beta}' \rangle, \quad \text{(F.2)}$$

where $d^{2M} \boldsymbol{\alpha} = d^M \Re(\boldsymbol{\alpha}) d^M \Im(\boldsymbol{\alpha})$. Using the expressions for the $Q$-functions of $\mathcal{U}_1$ and $\mathcal{U}_2$ we find

$$\frac{1}{\pi^M} \int_{\mathbb{C}^M} d^{2M} \boldsymbol{\alpha} \exp\left(-\tfrac{1}{2}\left[||\boldsymbol{\beta}||^2 + 2||\boldsymbol{\alpha}||^2 + ||\boldsymbol{\beta}'||^2\right]\right) c_{U_1} c_{U_2} \exp\left(b_{U_1}^T \boldsymbol{\nu_1} + \frac{1}{2} \boldsymbol{\nu}_1^T A_{U_1} \boldsymbol{\nu_1} + b_{U_2}^T \boldsymbol{\nu_2} + \frac{1}{2} \boldsymbol{\nu}_2^T A_{U_2} \boldsymbol{\nu_2}\right), \quad \text{(F.3)}$$

where

$$\boldsymbol{\nu}_1^T = [\boldsymbol{\beta}, \boldsymbol{\alpha}], \quad \boldsymbol{\nu}_2^T = [\boldsymbol{\alpha}^*, \boldsymbol{\beta}']. \quad \text{(F.4)}$$

The integral above can be simplified as

$$\frac{1}{\pi^M} c_{U_1} c_{U_2} \exp\left(-\tfrac{1}{2}\left[||\boldsymbol{\beta}||^2 + ||\boldsymbol{\beta}'||^2\right] + c_1^T \boldsymbol{\beta} + d_2^T \boldsymbol{\beta}' + \boldsymbol{\beta}^T B_1 \boldsymbol{\beta} + \boldsymbol{\beta}'^T D_2 \boldsymbol{\beta}'\right) \quad \text{(F.5)}$$

$$\times \int_{\mathbb{C}^M} d^{2M} \boldsymbol{\alpha} \exp\left(-\tfrac{1}{2}[\boldsymbol{\alpha}^T, \boldsymbol{\alpha}^{T*}] \begin{bmatrix} -D_1 & \mathbb{1}_M \\ \mathbb{1}_M & -B_2 \end{bmatrix} \begin{bmatrix} \boldsymbol{\alpha} \\ \boldsymbol{\alpha}^* \end{bmatrix} + [d_1^T + \boldsymbol{\beta}^T C_1, c_2^T + \boldsymbol{\beta}'^T C_2^T] \begin{bmatrix} \boldsymbol{\alpha} \\ \boldsymbol{\alpha}^* \end{bmatrix}\right), \quad \text{(F.6)}$$

where we introduced

$$b_{U_i}^T = \left[c_i^T, d_i^T\right], \quad A_{U_i} = \left[\begin{array}{c|c} B_i & C_i \\ \hline C_i^T & D_i \end{array}\right]. \quad \text{(F.7)}$$

The last integral can be written explicitly as

$$\frac{\pi^M}{\sqrt{\det(\mathbb{1}_M - D_1 B_2)}} \exp\left(\tfrac{1}{2}[d_1^T + \boldsymbol{\beta}^T C_1, c_2^T + \boldsymbol{\beta}'^T C_2^T] \begin{bmatrix} -D_1 & \mathbb{1}_M \\ \mathbb{1}_M & -B_2 \end{bmatrix}^{-1} \begin{bmatrix} d_1 + C_1^T \boldsymbol{\beta} \\ c_2 + C_2 \boldsymbol{\beta}' \end{bmatrix}\right), \quad \text{(F.8)}$$

where we wrote $\begin{bmatrix} \boldsymbol{\alpha} \\ \boldsymbol{\alpha}^* \end{bmatrix} = \boldsymbol{M} \begin{bmatrix} \Re(\boldsymbol{\alpha}) \\ \Im(\boldsymbol{\alpha}) \end{bmatrix}$ with $\boldsymbol{M} = \begin{bmatrix} \mathbb{1}_M & i\mathbb{1}_M \\ \mathbb{1}_M & -i\mathbb{1}_M \end{bmatrix}$, used the well known real integral $\int_{\mathbb{R}^n} d^n x \exp\left(-\frac{1}{2} x^T A x + b^\intercal x\right) = \sqrt{\frac{(2\pi)^n}{\det A}} \exp\left(\frac{1}{2} b^T A^{-1} B\right)$ and the fact that $\det\left(\boldsymbol{M}^T \begin{bmatrix} -\boldsymbol{D}_1 & \mathbb{1}_M \\ \mathbb{1}_M & -\boldsymbol{B}_2 \end{bmatrix} \boldsymbol{M}\right) = 2^{2M} \det(\mathbb{1}_M - \boldsymbol{D}_1 \boldsymbol{B}_2)$ (see also Theorem 3 of Appendix A of Folland [21]).

If we define

$$\mathcal{Y} = \mathbb{1}_M - \boldsymbol{B}_2 \boldsymbol{D}_1 \text{ (note that } \mathcal{Y}^T = \mathbb{1}_M - \boldsymbol{D}_1 \boldsymbol{B}_2 \text{ since } \boldsymbol{D}_1, \boldsymbol{B}_2 \text{ are symmetric)}, \tag{F.9}$$

then the inverse appearing in the last equation can be obtained using Schur complements and is given by,

$$\mathcal{Z} = \mathcal{Z}^T = \begin{bmatrix} -\boldsymbol{D}_1 & \mathbb{1}_M \\ \mathbb{1}_M & -\boldsymbol{B}_2 \end{bmatrix}^{-1} = \begin{bmatrix} \mathcal{Y}^{-1} \boldsymbol{B}_2 & \mathcal{Y}^{-1} \\ [\mathcal{Y}^T]^{-1} & \boldsymbol{D}_1 \mathcal{Y}^{-1} \end{bmatrix}. \tag{F.10}$$

This expression allows us to write

$$\langle \boldsymbol{\beta}^* | \mathcal{U}_f | \boldsymbol{\beta}' \rangle = \frac{c_{U_1} c_{U_2}}{\sqrt{\det(\mathcal{Y})}} \exp\left(-\frac{1}{2}\left[||\boldsymbol{\beta}||^2 + ||\boldsymbol{\beta}'||^2\right] + \boldsymbol{c}_1^T \boldsymbol{\beta} + \boldsymbol{d}_2^T \boldsymbol{\beta}' + \boldsymbol{\beta}^T \boldsymbol{B}_1 \boldsymbol{\beta} + \boldsymbol{\beta}'^T \boldsymbol{D}_2 \boldsymbol{\beta}'\right)$$

$$\times \exp\left(\frac{1}{2}[\boldsymbol{d}_1^T + \boldsymbol{\beta}^T \boldsymbol{C}_1, \boldsymbol{c}_2^T + \boldsymbol{\beta}'^T \boldsymbol{C}_2^T] \mathcal{Z} \begin{bmatrix} \boldsymbol{d}_1 + \boldsymbol{C}_1^T \boldsymbol{\beta} \\ \boldsymbol{c}_2 + \boldsymbol{C}_2 \boldsymbol{\beta}' \end{bmatrix}\right) \tag{F.11}$$

$$= \exp\left(-\frac{1}{2}\left[||\boldsymbol{\beta}||^2 + ||\boldsymbol{\beta}'||^2\right]\right) \underbrace{\frac{c_{U_1} c_{U_2}}{\sqrt{\det(\mathcal{Y})}} \exp\left(\frac{1}{2}[\boldsymbol{d}_1^T, \boldsymbol{c}_2^T] \mathcal{Z} \begin{bmatrix} \boldsymbol{d}_1 \\ \boldsymbol{c}_2 \end{bmatrix}\right)}_{\equiv c_{U_f}}$$

$$\times \exp\left(\underbrace{\left[[\boldsymbol{c}_1^T, \boldsymbol{d}_2^T] + [\boldsymbol{d}_1^T, \boldsymbol{c}_2^T] \mathcal{Z} \left\{\boldsymbol{C}_1^T \oplus \boldsymbol{C}_2\right\}\right]}_{\equiv b_{U_f}^T} \begin{bmatrix} \boldsymbol{\beta} \\ \boldsymbol{\beta}' \end{bmatrix}\right)$$

$$\times \exp\left(\frac{1}{2}[\boldsymbol{\beta}^T, \boldsymbol{\beta}'^T] \underbrace{\left[\boldsymbol{B}_1 \oplus \boldsymbol{D}_2 + \left\{\boldsymbol{C}_1 \oplus \boldsymbol{C}_2^T\right\} \mathcal{Z} \left\{\boldsymbol{C}_1^T \oplus \boldsymbol{C}_2\right\}\right]}_{\equiv A_{U_f}} \begin{bmatrix} \boldsymbol{\beta} \\ \boldsymbol{\beta}' \end{bmatrix}\right). \tag{F.12}$$

## G  Riemannian gradient of the unitary group

The Riemannian metric of the unitary group at point $\boldsymbol{A}$ is:

$$\langle \boldsymbol{X}, \boldsymbol{Y} \rangle_{\boldsymbol{A}} = \langle \boldsymbol{A}^{-1} \boldsymbol{X}, \boldsymbol{A}^{-1} \boldsymbol{Y} \rangle_{\mathbb{1}_{2n}} = \text{Tr}\left((\boldsymbol{A}^{-1} \boldsymbol{X})^\dagger \boldsymbol{A}^{-1} \boldsymbol{Y}\right), \quad \boldsymbol{X}, \boldsymbol{Y} \in T_{\boldsymbol{A}} \text{U}(n, \mathbb{C}). \tag{G.1}$$

The Riemannian gradient $\nabla_{\boldsymbol{A}} f$ at point $\boldsymbol{A}$ of a sufficiently regular function $f : \text{U}(n, \mathbb{C} \to \mathbb{C})$ associated to the Riemannian metric satisfies

$$\nabla_{\boldsymbol{A}} f = \frac{1}{2}\left(\partial_{\boldsymbol{A}} f - \boldsymbol{A} \partial_{\boldsymbol{A}}^\dagger f \boldsymbol{A}\right). \tag{G.2}$$

*Proof.* According to the compatibility of the Riemannian gradient with the Riemannian metric (defined in Eq. (G.1)), we have:

$$\langle \nabla_{\boldsymbol{A}} f, \boldsymbol{T} \rangle_{\boldsymbol{A}} = \langle \partial_{\boldsymbol{A}} f, \boldsymbol{T} \rangle_{\text{euc}}, \quad \forall \boldsymbol{T} \in T_{\boldsymbol{A}} \text{Sp}, \tag{G.3}$$

that it,

$$\langle \partial_A f - A^{-\dagger} A^{-1} \nabla_A f, T \rangle_{\text{euc}} = 0 \,. \tag{G.4}$$

This implies that $\partial_A f - A^{-\dagger} A^{-1} \nabla_A f \in N_A U$. So we have, with $AA^{\dagger} = \mathbb{1}$:

$$\partial_A f - A^{-\dagger} A^{-1} \nabla_A f = AN \,, \tag{G.5}$$

$$\partial_A f = \nabla_A f + AN \,. \tag{G.6}$$

Using the tangency condition $\nabla_A f \in T_A U$, we know

$$(\nabla_A f)^{\dagger} A + A^{\dagger} \nabla_A f = 0_n \,. \tag{G.7}$$

Together Eq. (G.7) and Eq. (G.6) with $N = N^{\dagger}$, we solve

$$N = \frac{1}{2} \left( A^{\dagger} \partial_A f + \partial_A^{\dagger} f A \right) \,. \tag{G.8}$$

Thus we obtain

$$\nabla_A f = \partial_A f - A \frac{1}{2} \left( A^{\dagger} \partial_A f + \partial_A^{\dagger} f A \right) \tag{G.9}$$

$$= \frac{1}{2} \left( \partial_A f - A \partial_A^{\dagger} f A \right) \,. \tag{G.10}$$

$\square$

# H  Solution of the cubic phase state optimization

Here we report the symplectic matrix of the Gaussian transformation in circuit 4b:

```
[[ 0.336437829, -0.587437101,  0.151967502,  2.011467789,  1.858626268, -1.401857238],
 [ 1.416888301,  0.409496273,  0.448704546, -1.759418716, -5.552019032,  2.056880833],
 [-0.477864655,  0.14143573,  -2.111321823, -2.485020087, -4.623168982,  2.511539347],
 [-5.701053833,  1.587452315,  0.364136769, -1.343878855, 12.237643127, -2.543280972],
 [-2.302558433,  1.344598162,  0.378523959, -2.291630056,  3.35733036 ,  0.527469667],
 [-1.386435201,  0.479622105, -0.771833605, -1.523680547,  0.579084776,  0.246557173]]
```

As well as the displacements in $x$ and $y$ of the three displacement gates:

```
[-0.642981239,  2.326888363,  3.021233284]
[ 2.266497837, -1.655566694, -2.858640664]
```

Note that we formatted them so that they can be easily copy-pasted from this document to a Jupyter notebook, or Python file (for instance to create a Numpy array).

# I  Comparison of Riemannian optimization and Euclidean optimization

We compare different optimization circuits to show the fast convergence of the Riemannian method. The 4-mode circuit device has three different structures shown in Fig. 8, which include three optimization methods:

(a) Symplectic optimization, in which the symplectic matrix of a Gaussian multimode gate is updated following the Riemannian manifold gradient as in Eq. (138).

(b) Unitary optimization, in which we update the unitary matrix of the Interferometer on its Riemannian manifold as in Eq. (141). The squeezing gates are optimized using Euclidean gradients.

(c) Euclidean optimization, in which all the parameters of a gate-decomposed circuit are updated according to the gradient.

Note that there are several methods to decompose a multi-mode interferometer into a beam-splitter network (of which many implementations exist [71–75]) and here we just pick one of them (shown in the circuit (c) of Fig. 8) to show the optimization.

We performed three experiments with a randomly chosen PNR pattern $[3, 1, 5]$ on the last three modes. The cost function to be optimized is the fidelity between the output state and the target cat state. The learning rate is chosen as 0.003 for the three optimization methods. This is an important hyperparameter and there is a significant room for further investigation, however for the sake of fairness we choose the same value for all three of them.

The results are shown in Fig. 9, which represents the average of the loss curves with 20 random seed optimization processes. The green solid line shows that circuit (a) in Fig. 8 has the fastest convergence achieving unit fidelity in around 10 steps. Surprisingly, the blue solid line shows that circuit (b) is slower than the orange solid line corresponding to circuit (c).

We analyze the optimization timing distribution of each run in Fig. 10. Two Riemannian methods (symplectic and unitary group optimization) spend around 1.25 seconds for each optimization, which is two times faster compared with the Euclidean method which finished in around 2.5 seconds.

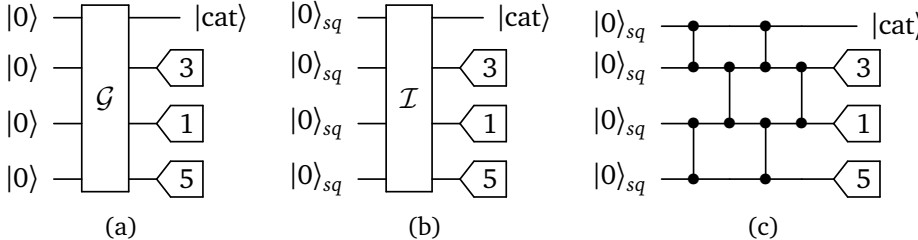

Figure 8: Four-mode circuit optimization. (a) Four-mode Gaussian transformation with PNR on the last three modes. This shows the symplectic optimization of the Gaussian object. (b) Four single-mode squeezed vacuum followed by a four-mode Interferometer with PNR on the last three modes. This includes both Euclidean optimization of the four squeezed vacuum states and unitary optimization of the interferometer. (c) Four single-mode squeezed vacuum followed by a beamsplitter network on every two modes with PNR on the last three modes. This includes both Euclidean optimization of the four squeezed vacuum states and the beamsplitters.

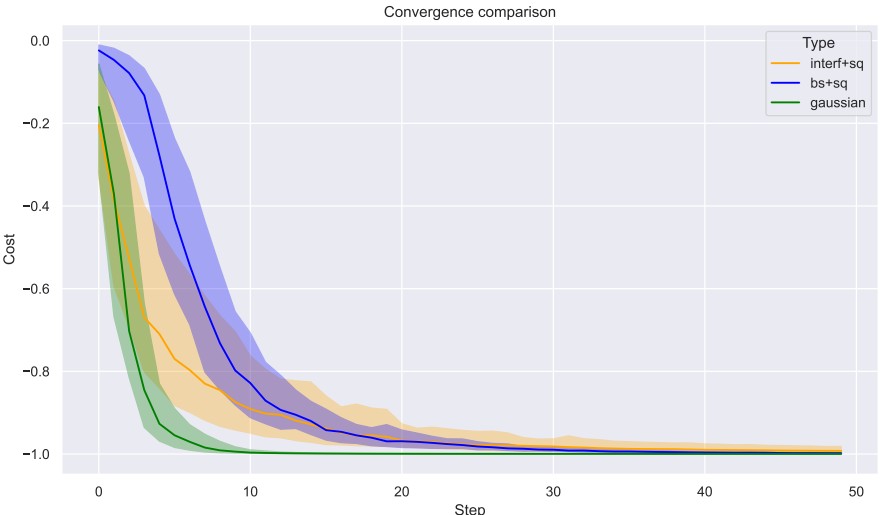

Figure 9: Comparison of the cost function in steps for three optimization circuits to prepare the cat state. The solid line represents the average of cost functions with 20 random seeds. The green solid line corresponds to circuit (a) in Fig. 8, the orange line corresponds to circuit (c) and the blue one corresponds to circuit (b).

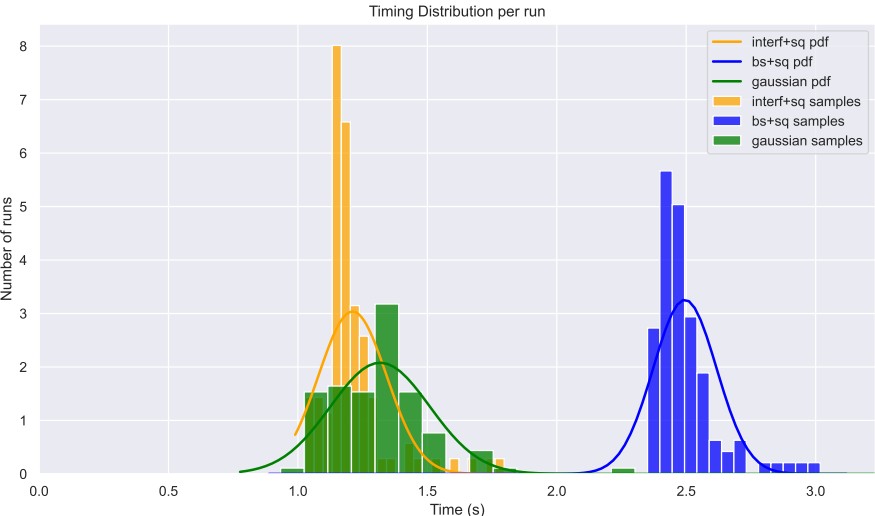

Figure 10: Time distribution of the training. The green line group corresponds to circuit (a) in Fig. 8, the orange line group corresponds to circuit (c) and the blue line group corresponds to circuit (b). In each group, the probability distribution function and the sample distribution are plotted.

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
