# Peer review of "Riemannian optimization of photonic quantum circuits in phase and Fock space"

_SciPost Physics, doi:SciPost Phys. 17, 082 (2024)_

## Round 4 · Referee Report · Anonymous (Referee 1) · 2023-11-12

Report

The manuscript provides a method to optimize quantum circuits based on Gaussian processes. This optimization is rendered possible thanks to a linear recurrence relation connecting the Fock and the Gaussian (Wigner) representations of quantum states, unitaries and channel. Specifically, optimization over Gaussian processes of a cost function in the Fock space is done by the backpropagation of a gradient from the Fock representation to the Gaussian representation and back. At the core of the presented method is a Riemannian optimization ; the authors derived a geodesic updating formula allowing to update symplectic matrices (which are the object representing Gaussian processes) along a geodesic path and according to a Riemannian gradient that can be computed from the cost function.
To display the proposed optimization methods, the authors provide three numerical experiments. They are performed using an open-source library which implements the optimization, thanks to contributions from the authors. Numerical results display convergence in the optimization of non-trivial tasks, therefore emphasising the optimization capability of both the proposed method and its implementation.

The method presented in this manuscript will appeal to readers working with photonic circuits, as I can think of multiple use-cases in which having a performant and efficient Gaussian and non-Gaussian (photodetectors) processes optimization framework would be highly appreciated. Furthermore, the Python package is of high-quality, from its simplicity of use to its documentation.
In my opinion, however, the discussion section is missing some important elements, contrasting the optimization method against existing ones, and mentioning potential limits/drawbacks of the approach (cf. Requested Changes).

Given the quality of this manuscript, the usefulness and novelty of the proposed framework, and provided that the authors answer the points mentioned in the requested changes section, I will support the publication of this manuscript in SciPost.

Requested changes

  1. Section "V-F. Discussion"

As a general remark, the quality of this subsection does not match the rest of the manuscript. I would therefore suggest a rewrite of this section.

More specifically, here are some keys points I would like to see included in the discussion:

a. The authors refer to a missing Appendix, containing convergences results. Such results are more than welcome, the new manuscript version should therefore include them. b. A comparison against different methods for the optimisation of general Gaussian processes would be appreciated. For example, how much faster does this method converge vs a black-box optimisation in the case where the objective function is to find a process matching (fidelity) a random unitary? Similar question if you optimise Gaussian components individually vs in bulk? c. What is the scalability of this method? For example, given the task of maximizing the fidelity to a randomly generated unitary on n-bosonic modes, how does the convergence-time scale with n? d. I would like to read more about the potential drawbacks and limits of this method. For example: Are there cases in which the optimisation is unstable/unreliable? What is the main computational bottleneck ? The authors mention some occurring “inaccuracies”, where do they originate from (renormalization?), and can we detect them to make the optimisation more trustworthy?

2 - References and Bibliography

A lot of the references in the bibliography are not with clickable URLs. On a side note, I also suggest these references to the authors:

A review on Gaussian quantum information - 10.1103/RevModPhys.84.621 A framework in which heralded Gaussian states are simulated using linear combination of Gaussian states - 10.1103/PhysRevA.107.062607 A quick references to common Gaussian states and operations - arXiv:2102.05748 Quantum Optics book in which quantum system in phase space are described - 10.1002/3527608524 A framework with auto-differentiation of optical setup processes - arXiv:2310.08408

3 - Editing

Some suggestions regarding typos or English formulation:

p.5: "k + 1i is like k but the i-th index has been increased by 1" -> "k +1i is the vector k with the i-th element increased by 1.". p.6: "and then we can write" -> "allowing us to write" . p.12: "its last two modes:" -> replace the column by a dot.

---

## Round 4 · Referee Report · Anonymous (Referee 2) · 2023-12-17

Report

The paper "On the design of photonic quantum circuits" presents a framework for designing and optimizing photonic quantum circuits involving Gaussian states, unitaries, channels, and non-Gaussian effects.

The main feat is the coherent introduction of a recurrent relation to define a phase space representation of Gaussian objects using Fock representation. The method comes with a description of state-of-the-art optimization techniques of quantum circuits and specifically their photonic implementation, such as Riemann optimization, and high-performance computational techniques such as auto-differentiation. Probably the most important and useful part of this paper is that it comes with an open-source library MrMustard. I have already played with the library (before I was asked to review this paper), and it is clearly of very high quality and very intuitive to use. Importantly, and this is why I believe that the software as well as this paper will have of huge impact: The software is continuously expanded (the last update on GitHub was 3 days ago, and it counts 18 contributors).

The manuscript is an extended documentation which describes the underlying mathematics of the software. As such, it will likely become a standard text for newcomers in the field who want to learn about the details of Gaussian quantum optics.

I think they should certainly be published, but I would like the authors to consider these points before they submit a final version:

1) I see that the authors changed the manuscript title on arxiv between versions 3 and 4 from 'The recursive representation of Gaussian quantum mechanics' to 'On the design of photonic quantum circuits'. I believe that the current title does not well describe the content of the paper. After all, there are by now dozens of papers that perform designs of photonic quantum circuits. I recommend to fix and make it clear what the content is about.

2) Furthermore, while computer-designed quantum photonic circuits are heavily studied since around 2016 (10.1088/1367-2630/18/7/073033, 10.1103/PhysRevLett.116.090405, 10.1088/2058-9565/aaf59e, https://github.com/Budapest-Quantum-Computing-Group/piquasso, https://github.com/xvalcarce/QuantumOpticalCircuits.jl used in 10.1103/PhysRevA.107.062607 and many others), the authors do not cite any other works in this field [except other Xanadu paper in ref 18 and 19, QuTiP which has a different purpose and a very recent 2022 Perceval software, and later some graph-based approaches in ref45]. I recommend comparing and contrasting their methods with currently existing techniques and explaining their applicability. As the authors have the "design" in the title, the comparison should talk about other design approaches (not just simulators).

3) The algorithm performs high-performance continuous optimization. However, is it possible to perform discrete optimization, i.e. starting with a circuit ansatz and not only finding the correct parameters but trying to reduce the circuit to smaller component numbers?

4) [optional] It would be interesting to give an overview of other fields that apply auto-differentiation. For example, what distinguishes the authors' methods from classical optics design methods in the field of photonic material design (e.g. 10.1021/acsphotonics.0c00327). This field is powered by the adjoint method, which is a special case of auto-differentiation.

---

## Round 5 · Referee Report · Mario Krenn (Referee 3) · 2024-7-13

Report

I read the response to my previous questions, and the modified pieces in the manuscript. As pointed out in my previous referee report, the software is highly useful, and this physical documentation will be of value to many working in the field of gaussian photonic states.

I strongly recommend the publication of this manuscript.

Recommendation

Publish (easily meets expectations and criteria for this Journal; among top 50%)

---

## Round 5 · Referee Report · Anonymous (Referee 4) · 2024-7-16

Report

The authors have answered all the points raised in my previous report. The updated manuscript includes new elements answering my questions and framing the work in a clear context. This is particularly the case for the newly added Appendix I on convergence speed, and the reworked Discussion section (V.F.).

From the quality of the manuscript and the usefulness of the proposed framework, I strongly recommend its publication in SciPost.

Recommendation

Publish (easily meets expectations and criteria for this Journal; among top 50%)

---

## Round 5 · Author Response

Reply on the paper "On the design of photonic quantum circuits"
We thank the editor and the reviewers for their effort and time going through our manuscript.
Below we provide detailed replies to the comments raised by the reviewers detailing the modifications we undertook based on their feedback.
We would like to highlight that both reviewers support the publication of the manuscript (provided some changes are performed). It is our hope this new version contains the necessary revision and thus will be suitable for publication in SciPost Physics.

---

## Round 5 · List of Changes

Below we reply to to the queries from the reviewers. Our replies are written between two asterisks, **like this**.

Referee 1:
The paper "On the design of photonic quantum circuits" presents a framework for designing and optimizing photonic quantum circuits involving Gaussian states, unitaries, channels, and non-Gaussian effects.
The main feat is the coherent introduction of a recurrent relation to define a phase space representation of Gaussian objects using Fock representation. The method comes with a description of state-of-the-art optimization techniques of quantum circuits and specifically their photonic implementation, such as Riemann optimization, and high-performance computational techniques such as auto-differentiation. Probably the most important and useful part of this paper is that it comes with an open-source library MrMustard. I have already played with the library (before I was asked to review this paper), and it is clearly of very high quality and very intuitive to use. Importantly, and this is why I believe that the software as well as this paper will have of huge impact: The software is continuously expanded (the last update on GitHub was 3 days ago, and it counts 18 contributors).
The manuscript is an extended documentation which describes the underlying mathematics of the software. As such, it will likely become a standard text for newcomers in the field who want to learn about the details of Gaussian quantum optics.
I think they should certainly be published, but I would like the authors to consider these points before they submit a final version:
**We thank the reviewer for recognizing the value of our work both in this manuscript and in the software library and for supporting the publication of our work. As the reviewer points out, we continue contributing to the library, adding new features based on the theory we developed in this manuscript.**

1) I see that the authors changed the manuscript title on arxiv between versions 3 and 4 from 'The recursive representation of Gaussian quantum mechanics' to 'On the design of photonic quantum circuits'. I believe that the current title does not well describe the content of the paper. After all, there are by now dozens of papers that perform designs of photonic quantum circuits. I recommend to fix and make it clear what the content is about.
**We thank the reviewer for raising this point. Indeed, we did ourselves a disservice by using such a generic title. We have modified the title in the hope that it better reflects the work in it.**
2) Furthermore, while computer-designed quantum photonic circuits are heavily studied since around 2016 (10.1088/1367-2630/18/7/073033, 10.1103/PhysRevLett.116.090405, 10.1088/2058-9565/aaf59e, https://github.com/Budapest-Quantum-Computing-Group/piquasso, https://github.com/xvalcarce/QuantumOpticalCircuits.jl used in 10.1103/PhysRevA.107.062607 and many others), the authors do not cite any other works in this field [except other Xanadu paper in ref 18 and 19, QuTiP which has a different purpose and a very recent 2022 Perceval software, and later some graph-based approaches in ref45]. I recommend comparing and contrasting their methods with currently existing techniques and explaining their applicability. As the authors have the "design" in the title, the comparison should talk about other design approaches (not just simulators).
**We thank the reviewer for bringing to our attention the libraries mentioned above. We now cite Piquasso (whose arXiv preprint only appeared in March of this year) as Ref. [24] and QuantumOpticalCircuits.jl as Ref. [25]. We have also further clarified in the introduction what sets our software library apart, namely “ours is the first one to fully exploit the properties of Gaussian quantum mechanics in Fock- and Phase-space while being differentiable.” to provide a precise definition what differentiates our work from the one of other groups.**
3) The algorithm performs high-performance continuous optimization. However, is it possible to perform discrete optimization, i.e. starting with a circuit ansatz and not only finding the correct parameters but trying to reduce the circuit to smaller component numbers?
**As highlighted in a new appendix (and further elaborated in the Discussion, Sec. V.F.) one of the merits of our library is that the whole space of Gaussian circuits can be parametrized directly without requiring gate decompositions. One can of course use particular arrangements of gates, as we highlight in Appendix I where we decompose circuits into passive plus active gates (and we do Riemannian optimization on the former) or also where we decompose the whole circuit in terms of single- and two-mode gates. **

4) [optional] It would be interesting to give an overview of other fields that apply auto-differentiation. For example, what distinguishes the authors' methods from classical optics design methods in the field of photonic material design (e.g. 10.1021/acsphotonics.0c00327). This field is powered by the adjoint method, which is a special case of auto-differentiation.
**We thank the referee for this question. We would like to emphasize that ours is a method for the design of quantum circuits, but we make no attempt to find ways to physically measure gradients. In fact, this is likely a hard question as unlike for qubits, the unitary representations of the symplectic group are non compact. So the usual “parameter shift rules” that have huge success in the qubit or finite dimensional world will likely not carry over immediately due to the presence of hyperbolic functions rather than trigonometric functions.**

Referee 2:
Report
The manuscript provides a method to optimize quantum circuits based on Gaussian processes. This optimization is rendered possible thanks to a linear recurrence relation connecting the Fock and the Gaussian (Wigner) representations of quantum states, unitaries and channel. Specifically, optimization over Gaussian processes of a cost function in the Fock space is done by the backpropagation of a gradient from the Fock representation to the Gaussian representation and back. At the core of the presented method is a Riemannian optimization ; the authors derived a geodesic updating formula allowing to update symplectic matrices (which are the object representing Gaussian processes) along a geodesic path and according to a Riemannian gradient that can be computed from the cost function.
To display the proposed optimization methods, the authors provide three numerical experiments. They are performed using an open-source library which implements the optimization, thanks to contributions from the authors. Numerical results display convergence in the optimization of non-trivial tasks, therefore emphasising the optimization capability of both the proposed method and its implementation.
The method presented in this manuscript will appeal to readers working with photonic circuits, as I can think of multiple use-cases in which having a performant and efficient Gaussian and non-Gaussian (photodetectors) processes optimization framework would be highly appreciated. Furthermore, the Python package is of high-quality, from its simplicity of use to its documentation.
In my opinion, however, the discussion section is missing some important elements, contrasting the optimization method against existing ones, and mentioning potential limits/drawbacks of the approach (cf. Requested Changes).
Given the quality of this manuscript, the usefulness and novelty of the proposed framework, and provided that the authors answer the points mentioned in the requested changes section, I will support the publication of this manuscript in SciPost.
**We thank the referee for acknowledging the novelty and usefulness of our work and supporting the publication of our manuscript in SciPost.**

Requested changes
1. Section "V-F. Discussion"

As a general remark, the quality of this subsection does not match the rest of the manuscript. I would therefore suggest a rewrite of this section.

More specifically, here are some keys points I would like to see included in the discussion:

a. The authors refer to a missing Appendix, containing convergences results. Such results are more than welcome, the new manuscript version should therefore include them.
**We thank the reviewer for this comment. The promised appendix is now included in this updated version as appendix “i”; moreover, based on the reviewers comments, we have rewritten Sec. V.F. corresponding to the discussion.**

b. A comparison against different methods for the optimisation of general Gaussian processes would be appreciated. For example, how much faster does this method converge vs a black-box optimisation in the case where the objective function is to find a process matching (fidelity) a random unitary? Similar question if you optimise Gaussian components individually vs in bulk?
**Thanks for suggesting this comparison. We have now performed a study of different *gradient* based methods involving Euclidean and Riemannian optimization. These results further back our claim that performing Riemannian optimization on the symplectic manifold indeed converges faster to (local) minima.**

c. What is the scalability of this method? For example, given the task of maximizing the fidelity to a randomly generated unitary on n-bosonic modes, how does the convergence-time scale with n?
**This is an excellent question. The complexity will depend on how many modes are mapped into Fock space. If everything is kept in phase-space, as it is well known, the complexity is polynomial and roughly equivalent to matrix multiplication; this allows us to minimize the sparsity of 216 mode high-dimensional GBS instance in Sec. VI. For the case when one has to leave Fock space and is interested in heralding 1 out of N modes, significant speedups can be obtained by exploiting the symmetries of the recursion relations. This is explored in detail in the new Ref. 47 of the update manuscript. We also highlight this speed up as a new sentence at the end of Sec. III.A. **

d. I would like to read more about the potential drawbacks and limits of this method. For example: Are there cases in which the optimisation is unstable/unreliable? What is the main computational bottleneck ? The authors mention some occurring “inaccuracies”, where do they originate from (renormalization?), and can we detect them to make the optimisation more trustworthy?
We thank the referee for this excellent point. Indeed, when going to high cutoffs (of around 100) catastrophic loss of precision can happen. To solve this problem, MrMustard has been reimplemented using high-precision data-types that allows us to mitigate this issue (at the expense of longer calculation times).

2 - References and Bibliography

A lot of the references in the bibliography are not with clickable URLs.
On a side note, I also suggest these references to the authors:

A review on Gaussian quantum information - 10.1103/RevModPhys.84.621
A framework in which heralded Gaussian states are simulated using linear combination of Gaussian states - 10.1103/PhysRevA.107.062607
A quick references to common Gaussian states and operations - arXiv:2102.05748
Quantum Optics book in which quantum system in phase space are described - 10.1002/3527608524
A framework with auto-differentiation of optical setup processes - arXiv:2310.08408
**Thanks to the reviewer for suggesting these references. We now cite “A framework in which heralded Gaussian states are simulated using linear combination of Gaussian states” as Ref. [30] in the intro when we talk about other libraries. We cite “A review on Gaussian quantum information” as Ref. [2], “A quick references to common Gaussian states and operations” as Ref. [3] and “Quantum Optics book in which quantum system in phase space are described” Ref. [4] in the intro when we talk about Gaussian quantum mechanics.**

---

## Editorial Decision

published